# Mirror Descent with Relative Smoothness in Measure Spaces, with application to Sinkhorn and EM

**Pierre-Cyril Aubin-Frankowski**
DI ENS, Ecole normale supérieure,
Université PSL, CNRS, INRIA Paris
pierre-cyril.aubin@inria.fr

**Anna Korba**
CREST, ENSAE
IP Paris
anna.korba@ensae.fr

**Flavien Léger**
INRIA Paris
flavien.leger@inria.fr

## Abstract

Many problems in machine learning can be formulated as optimizing a convex functional over a vector space of measures. This paper studies the convergence of the mirror descent algorithm in this infinite-dimensional setting. Defining Bregman divergences through directional derivatives, we derive the convergence of the scheme for relatively smooth and convex pairs of functionals. Such assumptions allow to handle non-smooth functionals such as the Kullback–Leibler (KL) divergence. Applying our result to joint distributions and KL, we show that Sinkhorn's primal iterations for entropic optimal transport in the continuous setting correspond to a mirror descent, and we obtain a new proof of its (sub)linear convergence. We also show that Expectation Maximization (EM) can always formally be written as a mirror descent. When optimizing only on the latent distribution while fixing the mixtures parameters – which corresponds to the Richardson–Lucy deconvolution scheme in signal processing – we derive sublinear rates of convergence.

## 1 Introduction

Many important problems in machine learning and computational statistics can be cast as an optimization problem over the space of probability distributions, where the objective functional assesses the dissimilarity to a target distribution $\bar{\mu}$ on $\mathbb{R}^d$. Classical dissimilarities include $f$-divergences, Integral Probability Metrics (IPMs), or optimal transport distances among others. In Bayesian inference for instance, it is common to optimize the Kullback–Leibler (KL) divergence to the target, which corresponds to the posterior distribution of the parameters of interest. In generative modelling, the goal is to generate data whose distribution is similar to the training set distribution defined by samples of the target, where the similarity is often measured by an integral probability metric or an optimal transport distance (Arjovsky et al., 2017; Dziugaite et al., 2015). In supervised learning, optimizing an infinite-width one hidden layer neural network, e.g. through the mean squared error, corresponds to minimizing a functional on the space of probability distributions over the parameters of the network (Chizat and Bach, 2018; Mei et al., 2018; Rotskoff and Vanden-Eijnden, 2018). In particular, the objective functional can be identified to a Maximum Mean Discrepancy (MMD) in the well-specified setting (Arbel et al., 2019). Many other problems in machine learning can be formalized in this framework (Chu et al., 2019).

Once the objective functional is chosen, one has to select an optimization algorithm that is well-suited to the geometry of the problem. In this article, we consider the widely used mirror descent scheme, a first-order optimization method based on Bregman divergences. While the traditional

smoothness and strong convexity assumptions, required by standard convergence analysis, do not always hold over measure spaces, their "relative" versions have received increased interest. The relative smoothness assumption was first suggested by Birnbaum et al. (2011) in the context of algorithmic game theory, but remained unnoticed by the optimization community, until Bauschke et al. (2017) discovered the same concept independently, while Lu et al. (2018) coupled it with relative strong convexity. We extend their work to the infinite dimensional setting and target specifically the KL divergence. Indeed, when using the entropy as Bregman divergence, mirror descent is known to yield multiplicative updates for the measures. In this paper we study two such schemes in machine learning, i.e. Sinkhorn's algorithm, widely used to solve entropic optimal transport (Peyré and Cuturi, 2019), and the EM algorithm, a very common approach to fitting probabilistic models.

**Related work.** Chizat (2021) gave convergence rates of mirror descent on measure spaces for integral functionals with Lipschitz gradients, working mostly in the $L^1$ space and without leveraging relative smoothness. However his assumptions do not cover $f$-divergences, such as the ubiquitous KL, nor entropic regularized transport as in Léger (2020). His setting also implies Gâteaux differentiability, which is classical for mirror descent in Banach spaces (Bauschke et al., 2003). As the entropy is not differentiable in infinite dimensions, directional derivatives were instead used by Resmerita (2005) for mirror descent, and applied by Chu et al. (2019) to gradient descent. Connections between mirror descent and Sinkhorn iterations for the entropic optimal transport problem were first investigated in Mishchenko (2019); Mensch and Peyré (2020); Léger (2020). Our framework is closest to Léger (2020), which we simplify by considering primal rather than dual iterations, and extend by also deriving a linear convergence rate. Kunstner et al. (2021) recently showed that EM over parametric exponential mixtures corresponds to a mirror descent scheme with relative smoothness properties. Their setting is complementary to ours since we consider instead fixed mixtures and a nonparametric latent distribution. A prominent alternative setting to ours when optimizing over measures is based on (grid-free) Wasserstein gradient flows, which we do not cover here (see Appendix E for a discussion on the difference of geometries). A remarkable discussion on the optimization over measures with different geometries can be found in Trillos and Sanz-Alonso (2020).

**Contributions.** We propose a rigourous framework for the analysis of the infinite-dimensional version of mirror descent over measure spaces. In this setting, we recover the rate of convergence of mirror descent under relative smoothness and convexity, previously shown in finite dimensions. Defining Bregman divergences through directional derivatives, we are able to consider objective functionals over measures that are not smooth in the "standard" sense, but satisfy relative smoothness and/or convexity with respect to a Bregman divergence, e.g. KL. Focusing on optimization over joint distributions, we show that both Sinkhorn's primal iterations for entropic optimal transport in the continuous setting and EM can be written as a mirror descent. We then obtain a new proof of Sinkhorn's (sub)linear convergence. For EM, when optimizing on the latent distribution while fixing the mixtures, a choice which coincides with Richardson–Lucy deconvolution, we derive new sublinear rates of convergence.

This paper is organized as follows. Section 2 introduces the necessary background on derivatives in measure spaces and relative smoothness and convexity. Section 3 discusses the well-posedness of the mirror descent scheme and provides our proof of convergence adapted from Lu et al. (2018). Section 4 recovers the convergence of algorithms such as Sinkhorn's iterations and latent EM as special cases of mirror descent with relative smoothness and convexity.

## 2 Background and definitions

In this section, we set the mathematical framework in which we will rigourously reformulate relative smoothness and convexity on a space of measures.

**Notation.** Given a topological vector space $\mathcal{Y}$ with topology $\tau$, the domain $\mathrm{dom}(f)$ of an extended-valued function $f : \mathcal{Y} \to \mathbb{R} \cup \{\pm\infty\}$ is the set of points of $\mathcal{Y}$ where $f$ takes finite values. The function $f$ is said to be proper if $\mathrm{dom}(f)$ is non-empty and if $f$ never takes the value $-\infty$. It is $\tau$-lower semicontinuous (l.s.c.) if its sublevel sets are $\tau$-closed. We consider a dual pair $(\mathcal{Y}, \mathcal{Y}^*)$ with duality product $\langle \cdot, \cdot \rangle_{\mathcal{Y}^* \times \mathcal{Y}}$ which induces a $\mathcal{Y}^*$-weak topology on $\mathcal{Y}$ (see Aliprantis and Border, 2006). We write $\mathrm{Int}\, C$ for the interior of a set $C \subset \mathcal{Y}$. Let $\mathcal{X} \subset \mathbb{R}^d$, and fix a locally convex topological vector space of measures $(\mathcal{M}(\mathcal{X}), \tau)$, which could be for instance $L^1(\mathrm{d}\rho)$, $L^2(\mathrm{d}\rho)$ where $\rho$ is a reference measure, or the space of Radon measures $\mathcal{M}_r(\mathcal{X})$ with the total variation (TV) norm.

Fix $\mathcal{M}^*(\mathcal{X})$ a topological dual of $\mathcal{M}(\mathcal{X})$. For $\mu \in \mathcal{M}(\mathcal{X})$ and $f \in \mathcal{M}^*(\mathcal{X})$, we use the shorthand $\langle f, \mu \rangle = \langle f, \mu \rangle_{\mathcal{M}^*(\mathcal{X}) \times \mathcal{M}(\mathcal{X})}$, formally equal to $\int_\mathcal{X} f(x)\mu(dx)$. We denote by $\mathcal{P}(\mathcal{X})$ the subset of measures $\mu \in \mathcal{M}(X)$ with mass 1, and, for any $\mu, \nu \in \mathcal{M}(\mathcal{X})$, we write $\mu \ll \nu$ when $\mu$ is absolutely continuous w.r.t $\nu$, i.e. when it has a Radon–Nikodym derivative $d\mu/d\nu$.

Consider a convex functional $\mathcal{F} : \mathcal{M}(\mathcal{X}) \to \mathbb{R} \cup \{+\infty\}$ and the following minimization problem

$$\min_{\nu \in C} \mathcal{F}(\nu) \tag{1}$$

where $C \subset \mathcal{M}(\mathcal{X})$ is a convex set. To solve this convex optimization problem, a classical choice is to resort to a mirror descent scheme (see e.g. Beck and Teboulle, 2003). The latter is a first-order optimization scheme based on the knowledge of the "derivative" of the objective functional $\mathcal{F}$ at each iteration. The difficulty is to choose the appropriate notion of derivative. Indeed, Gâteaux and Fréchet derivatives have to be defined in every direction (see Appendix A), thus requiring that the points of differentiability belong to the interior of the domain $\mathrm{Int}(\mathrm{dom}(\mathcal{F}))$ of the functional $\mathcal{F}$ considered. In infinite dimensions, for functionals defined on positive measures, such as the negative entropy, $\mathrm{Int}(\mathrm{dom}(\mathcal{F}))$ is however empty[1]. Consequently, following Resmerita (2005), we favor a weaker notion, that of directional derivatives.[2] This comes at the price of manipulating $\pm\infty$ values but ensures that the considered derivatives are always well-defined for convex functionals. Besides, whenever the directional derivative is a linear function in a restricted set of directions, the notion of first variation that comes next will enable us to perform all the computations we need, as if the function was Gâteaux differentiable.

**Definition 1** (Directional derivative). If it exists, the *directional derivative* of $\mathcal{F} : \mathcal{M}(\mathcal{X}) \to \mathbb{R} \cup \{\pm\infty\}$ at a point $\nu \in \mathrm{dom}(\mathcal{F})$ in the direction $\mu \in \mathcal{M}(\mathcal{X})$ is defined as

$$d^+\mathcal{F}(\nu)(\mu) = \lim_{h \to 0^+} \frac{\mathcal{F}(\nu + h\mu) - \mathcal{F}(\nu)}{h}. \tag{2}$$

**Remark 1.** In particular, for convex and proper functions, $d^+\mathcal{F}(\nu)(\mu)$ exists and belongs to $\mathbb{R} \cup \{\pm\infty\}$ (Aliprantis and Border, 2006, Lemma 7.14). This is a direct consequence of the nondecreasingness of $\mathbb{R}_+^* \ni h \mapsto \frac{\mathcal{F}(\nu+h\mu)-\mathcal{F}(\nu)}{h}$ for any $\nu, \mu$ and convex $\mathcal{F}$. The monotonicity also entails that $d^+\mathcal{F}(\nu)(\mu) \leq \mathcal{F}(\nu + \mu) - \mathcal{F}(\nu) < \infty$ whenever $\nu$ and $\nu + \mu$ belong to $\mathrm{dom}(\mathcal{F})$ and that $d^+$ is a linear operation over the cone of convex functions. Note that $\mu \mapsto d^+\mathcal{F}(\nu)(\mu)$ is not always $\tau$-l.s.c. although it is always positively homogeneous, and, whenever $\mathcal{F}$ is convex, it is convex.

**Definition 2** (First variation). Let $\mathcal{F} : \mathcal{M}(\mathcal{X}) \to \mathbb{R} \cup \{+\infty\}$ be a functional and $C$ be a subset of $\mathcal{M}(\mathcal{X})$. If it exists, the *first variation* of $\mathcal{F}$ over $C$ evaluated at $\mu \in \mathrm{dom}(\mathcal{F}) \cap C$ is the element $\nabla_C \mathcal{F}(\mu) \in \mathcal{M}^*(\mathcal{X})$, unique up to orthogonal components to $\mathrm{span}(\mathrm{dom}(\mathcal{F}) \cap C - \mu)$, such that:

$$\langle \nabla_C \mathcal{F}(\mu), \xi \rangle = d^+\mathcal{F}(\mu)(\xi) \tag{3}$$

for all $\xi = \nu - \mu \in \mathcal{M}(\mathcal{X})$, where $\nu \in \mathrm{dom}(\mathcal{F}) \cap C$.

By Remark 1, we have that $d^+\mathcal{F}(\mu)(\xi) \in [-\infty, \infty)$, since $\nu, \mu \in \mathrm{dom}(\mathcal{F}) \cap C$. Naturally, if $\mathcal{F}$ has a Fréchet or Gâteaux derivative at $\mu$, which implies that $\mu \in \mathrm{Int}(\mathrm{dom}(\mathcal{F}))$, then it coincides with the first variation at $\mu$. Calling first variations the derivatives of functionals stems from the field of calculus of variations; a pragmatic approach when dealing with probability measures can be found in Santambrogio (2015, Definition 7.12), where $\nabla_{\mathcal{P}(\mathcal{X})} \mathcal{F}(\mu)$ is defined as a measurable function, we instead take it most in often in $L^\infty(\mathcal{X})$. We now introduce Bregman divergences over measures through directional derivatives.

**Definition 3.** (Bregman divergence) Let $\phi : \mathcal{M}(\mathcal{X}) \to \mathbb{R} \cup \{+\infty\}$ be a convex functional. For $\mu \in \mathrm{dom}(\phi)$, the $\phi$-*Bregman divergence* is defined for all $\nu \in \mathrm{dom}(\phi)$ by

$$D_\phi(\nu | \mu) = \phi(\nu) - \phi(\mu) - d^+\phi(\mu)(\nu - \mu) \in [0, +\infty], \tag{4}$$

and $+\infty$ elsewhere. The function $\phi$ is referred to as *the Bregman potential*.

---

[1]Intuitively if $\mathcal{X}$ contains an open set, then any positive measure can be infinitesimally perturbed to have negative values. For finite sets $\mathcal{X}$, this phenomenon does not occur, see also Example 2.

[2]In finite dimensions, Maddison et al. (2021) also defined Bregman divergences through directional derivatives, but under the stringent assumption of essentially smooth convex functions, an assumption which does not extend well to infinite dimensions.

In the previous definition, the restriction to $\mathrm{dom}(\phi)$ is necessary to avoid the substraction of infinite values. As a direct consequence of Lemma 12 in Appendix, a stricly convex $\phi$ entails that the Bregman divergence $D_\phi$ separates measures, i.e. $D_\phi(\nu|\mu) = 0$ if and only if $\nu = \mu$. Note that $D_\phi(\cdot|\mu)$ is a difference of convex functions, so it is not convex in general. Nevertheless the existence of a first variation (3) of $\phi$ over $C$, resulting in the last term in (4) being linear, is sufficient to ensure the convexity of the restriction of $D_\phi(\cdot|\mu)$ to $C$. Bregman divergences have useful immediate properties: since $d^+$ is a linear operation over convex functions, so is the Bregman divergence, i.e. for two convex $\phi, \psi$, $D_{\phi+\psi} = D_\phi + D_\psi$. Moreover, it is idempotent, as shown in the following lemma.

**Lemma 1** (Idempotence of Bregman divergence). Let $\phi : \mathcal{M}(\mathcal{X}) \to \mathbb{R} \cup \{+\infty\}$ be a convex functional. Assume that given $\xi \in \mathrm{dom}(\phi)$, the first variation $\nabla_C \phi(\xi)$ exists, then, for all $\mu, \nu \in C \cap \mathrm{dom}(\phi)$, $D_{D_\phi(\cdot|\xi)}(\nu|\mu) = D_\phi(\nu|\mu)$.

*Proof.* Since $\psi : \tilde{\mu} \mapsto -d^+\phi(\xi)(\tilde{\mu} - \xi) = -\langle \nabla_C \phi(\xi), \tilde{\mu} - \xi \rangle$ is convex over $C \cap \mathrm{dom}(\phi)$, we can apply the linearity of the Bregman divergence:

$$D_{D_\phi(\cdot|\xi)}(\nu|\mu) = D_\phi(\nu|\mu) + D_{-\phi(\xi)}(\nu|\mu) + D_\psi(\nu|\mu) = D_\phi(\nu|\mu),$$

since the Bregman divergence of a constant or of a linear form is null. $\square$

We are now ready to introduce the notions of relative smoothness (Bauschke et al., 2017) and convexity (Lu et al., 2018) of a functional w.r.t a Bregman potential.

**Definition 4.** (Relative smoothness and convexity) Let $\mathcal{F} : \mathcal{M}(\mathcal{X}) \to \mathbb{R} \cup \{+\infty\}$ be a convex proper functional. Given a scalar $L \geq 0$, we say that $\mathcal{F}$ is $L$-smooth relative to $\phi$ over $C$ if, for any $\mu, \nu \in \mathrm{dom}(\mathcal{F}) \cap \mathrm{dom}(\phi) \cap C$, we have

$$D_\mathcal{F}(\nu|\mu) = \mathcal{F}(\nu) - \mathcal{F}(\mu) - d^+\mathcal{F}(\mu)(\nu - \mu) \leq L D_\phi(\nu|\mu). \tag{5}$$

Conversely, we say that $\mathcal{F}$ is $l$-strongly convex relative to $\phi$ over $C$, for some scalar $l \geq 0$, if, for any $\mu, \nu \in \mathrm{dom}(\mathcal{F}) \cap \mathrm{dom}(\phi) \cap C$, we have

$$D_\mathcal{F}(\nu|\mu) \geq l D_\phi(\nu|\mu). \tag{6}$$

**Example 1** (L-smoothness). Choosing $\phi(\mu) = \|\mu\|^2_{\mathcal{M}(\mathcal{X})}$ the square norm on $\mathcal{M}(\mathcal{X})$ shows that relative smoothness extends the notion of $L$-smooth functionals (see for instance Chizat (2021, Lemma 3.1)), i.e. functionals with $L$-Lipschitz Gâteaux derivative, thus satisfying:

$$\mathcal{F}(\nu) - \mathcal{F}(\mu) - d^+\mathcal{F}(\mu)(\nu - \mu) \leq L\|\nu - \mu\|^2. \tag{7}$$

Notice that by Lemma 1, provided $\nabla_C \phi(\xi)$ is well-defined, a Bregman divergence objective $D_\phi(\cdot|\xi)$ is always 1-relatively smooth and strongly convex w.r.t. $\phi$. This we will heavily exploit for mirror descent schemes that involve the KL divergence both as an objective and Bregman divergence in Section 4. Interestingly, relative smoothness and convexity can be characterized in different, equivalent ways, see Lemma 13 in the Appendix. We now turn to the analysis of the mirror descent scheme using the above framework.

## 3 Mirror descent over measures and convergence

In the following, $\phi$ is assumed to be strictly convex. The relative smoothness assumption (5) of the convex objective functional $\mathcal{F}$ w.r.t. a Bregman potential $\phi$ suggests to minimize iteratively over $\nu \in C$ the function $\nu \mapsto \mathcal{F}(\mu) + d^+\mathcal{F}(\mu)(\nu - \mu) + L D_\phi(\nu|\mu)$, acting as an upper approximation of $\mathcal{F}(\nu)$. Starting from a given $\mu_0 \in \mathcal{M}(\mathcal{X})$, the mirror descent iterates are thus defined at each time $n \geq 0$ as

$$\mu_{n+1} = \underset{\nu \in C}{\mathrm{argmin}} \{d^+\mathcal{F}(\mu_n)(\nu - \mu_n) + L D_\phi(\nu|\mu_n)\}. \tag{8}$$

Let $\mathcal{R} \subset C$ be a given subset. As proven later in this section, sufficient conditions for the convergence of the scheme (8) are:

($\mathbf{A}_1$) (Existence) The sequence of iterates $(\mu_n)_{n \in \mathbb{N}}$ defined by (8) exist, belong to $\mathcal{R}$, and are unique.

($\mathbf{A}_2$) (Relative smoothness/convexity) For some $l, L \geq 0$, the functional $\mathcal{F}$ is $L$-smooth and $l$-strongly convex relative to $\phi$ as in Definition 4 for elements of $\mathcal{R}$.

(**A₃**) (Existence of first variation of $\phi$) For each $n \geq 0$, the first variation $\nabla_C \phi(\mu_n)$ exists.

These assumptions have to be verified on a case-by-case basis. In the simplest case, one can take $\mathcal{R} = C$. However the set $\mathcal{R}$ does not have to be convex (see Section 4). On the other hand, Assumption (**A'₁**) below, ensures that the iterates in (8) are well-defined, uniqueness resulting from the strict convexity of $\phi$.

(**A'₁**) (Lower semicontinuity and coercivity) (i) the set $C$ is $\tau$-closed in $\mathcal{M}(\mathcal{X})$, the functionals $\mathcal{G}_n(\cdot) := d^+\mathcal{F}(\mu_n)(\cdot - \mu_n)$ and $D_\phi(\cdot|\mu_n)$ are proper and $\tau$-l.s.c. when restricted to $C$, and the functional $\mathcal{G}_n + D_\phi(\cdot|\mu_n) + i_C{}^3$ has at least one $\tau$-compact sublevel set. (ii) For each $n \geq 0$, the first variations $\nabla_C \phi(\mu_n)$ exist. (iii) The iterates belong to $\mathcal{R}$.

By Attouch et al. (2014, Theorem 3.2.2), Assumption (**A'₁**) implies (**A₁**). Indeed, (i) $\tau$-lower semicontinuity and $\tau$-compactness guarantee the existence of minimizers of the objective (8), while (ii) the existence of $\nabla_C \phi(\mu_n)$ guarantees that $D_\phi(\cdot|\mu_n)$ is strictly convex (since $\phi$ is assumed strictly convex) hence unicity of the minimizer. Regarding Assumption (**A'₁**)(i), notice that $\mathcal{G}_n$ is proper and $\mathcal{M}^*(\mathcal{X})$-weak-l.s.c. as soon as $\mathcal{F}$ has a first variation at $\mu_n$, since in this case $\mathcal{G}_n$ is linear on $C$. We refer to Appendix B for more details on proving that Assumption (**A'₁**) holds for some $\mathcal{F}$, $\phi$ and $C$.

Mirror descent can also be defined through a subdifferential constraint if $\mathcal{F}$ also has first variations.

**Lemma 2** (Mirror descent dual iteration). If $\nabla_C \mathcal{F}(\mu_n)$ and $\nabla_C \phi(\mu_n)$ exist for all $n \geq 0$, then (8) is equivalent to

$$\nabla_C \phi(\mu_n) - \frac{1}{L}\nabla_C \mathcal{F}(\mu_n) \in \partial_C \phi(\mu_{n+1}) := \{p \,|\, \forall \nu \in C, \, \langle p, \nu - \mu_{n+1}\rangle \leq d^+\phi(\mu_{n+1})(\nu - \mu_{n+1})\} \tag{9}$$

Thus, if $\partial_C \phi(\mu_{n+1}) = \{\nabla_C \phi(\mu_{n+1})\}$, (8) corresponds to $\nabla_C \phi(\mu_{n+1}) - \nabla_C \phi(\mu_n) = -\frac{1}{L}\nabla\mathcal{F}(\mu_n)$.

*Proof.* The minimization (8) is equivalent to having, for all $\nu \in C$,

$$d^+\mathcal{F}(\mu_n)(\nu - \mu_n) + LD_\phi(\nu|\mu_n) \geq d^+\mathcal{F}(\mu_n)(\mu_{n+1} - \mu_n) + LD_\phi(\mu_{n+1}|\mu_n)$$
$$\langle \nabla_C \mathcal{F}(\mu_n) - L\nabla_C \phi(\mu_n), \nu - \mu_{n+1}\rangle + L(\phi(\nu) - \phi(\mu_{n+1})) \geq 0.$$

Take $\tilde\nu \in C$, set $\nu = \mu_{n+1} + t(\tilde\nu - \mu_{n+1})$ for $t \in [0,1]$. Taking the limit $t \to 0^+$ yields the result. $\square$

In the general case, as discussed in finite dimensions in Bauschke et al. (2017, Remark 3), one needs extra assumptions to justify that $\mu_{n+1}$ exists in (9), akin to the invertibility of $\nabla\phi$ or that $\phi$ is essentially smooth or of Legendre type. To avoid any restrictive assumption required to use (9), we stick with the minimal formulation (8) as was also done by Bauschke et al. (2017); Lu et al. (2018).

We now state a preliminary result, known as the "three-point inequality" or "Bregman proximal inequality" in the optimization literature (Chen and Teboulle, 1993, Lemma 3.2), (Lan et al., 2011, Lemma 1), useful to prove the convergence of the mirror descent scheme, similarly to Lu et al. (2018).

**Lemma 3** (Three-point inequality). Given $\mu \in \mathcal{M}(\mathcal{X})$ and some proper convex functional $\mathcal{G} : \mathcal{M}(\mathcal{X}) \to \mathbb{R} \cup \{+\infty\}$, if $\nabla_C \phi(\mu)$ exists, as well as $\bar\nu = \operatorname{argmin}_{\nu \in C}\{\mathcal{G}(\nu) + D_\phi(\nu|\mu)\}$, then for all $\nu \in C \cap \operatorname{dom}(\phi) \cap \operatorname{dom}(\mathcal{G})$:

$$\mathcal{G}(\nu) + D_\phi(\nu|\mu) \geq \mathcal{G}(\bar\nu) + D_\phi(\bar\nu|\mu) + D_\phi(\nu|\bar\nu). \tag{10}$$

*Proof.* The existence of $\nabla_C \phi(\mu)$ entails that $C \cap \operatorname{dom}(D_\phi(\cdot|\mu)) = C \cap \operatorname{dom}(\phi)$. Set $f(\cdot) = \mathcal{G}(\cdot) + D_\phi(\cdot|\mu)$. Then, by linearity of the Bregman divergence and Lemma 1, we obtain that, for any $\nu \in C \cap \operatorname{dom}(\phi) \cap \operatorname{dom}(\mathcal{G})$,

$$D_f(\nu|\bar\nu) = D_\mathcal{G}(\nu|\bar\nu) + D_{D_\phi(\cdot|\mu)}(\nu|\bar\nu) = D_\mathcal{G}(\nu|\bar\nu) + D_\phi(\nu|\bar\nu) \geq D_\phi(\nu|\bar\nu). \tag{11}$$

By optimality of $\bar\nu$, for all $\nu \in C$, $d^+(f)(\bar\nu)(\nu - \bar\nu) = \lim_{h\to 0^+}(f((1-h)\bar\nu + h\nu) - f(\bar\nu))/h \geq 0$; which is equivalent to $f(\nu) \geq f(\bar\nu) + D_f(\nu|\bar\nu)$. We conclude using (11) and the definition of $f$. $\square$

---

³$i_C$ denotes the indicator function of the set $C$, defined by $i_C(\mu) = 0$ if $\mu \in C$, $+\infty$ otherwise for any $\mu \in \mathcal{M}(\mathcal{X})$. Notice that $i_C$ being $\tau$-l.s.c. is equivalent to $C$ being $\tau$-closed in $\mathcal{M}(\mathcal{X})$.

The following theorem gives the rate of convergence of mirror descent for relatively smooth and convex pairs of functionals, and extends to infinite dimensions the convergence result of Lu et al. (2018, Theorem 3.1). Its proof can be found in Appendix F.1.

**Theorem 4** (Convergence rate). Assume that Assumptions $(\mathbf{A}_1)$, $(\mathbf{A}_2)$ and $(\mathbf{A}_3)$ hold. Consider the mirror descent scheme (8), then for all $n \geq 0$ and all $\nu \in \mathrm{dom}(\mathcal{F}) \cap \mathrm{dom}(\phi) \cap \mathcal{R}$, we have

$$\mathcal{F}(\mu_n) - \mathcal{F}(\nu) \leq \frac{lD_\phi(\nu|\mu_0)}{\left(1 + \frac{l}{L-l}\right)^n - 1} \leq \frac{L}{n} D_\phi(\nu|\mu_0), \tag{12}$$

where, in the case $l = 0$, the middle expression is defined in the limit as $l \to 0^+$.

**Remark 2** (About the proof of convergence). Our proof resembles the one of Lu et al. (2018), which also relies on a three-point inequality as stated in Lemma 3. However, the proof of the latter inequality in finite dimensions relies on a sum of subdifferentials formula, which is hard to verify for general functionals and in particular for the KL divergeence defined below (see also Remark 6 in Appendix). On the contrary, by working with directional derivatives and first variations, we circumvent most of the difficulties related to (sub)differentiability.

An important example is the one discussed below where $\phi$ is chosen to be the negative entropy $\phi_e$.

**Example 2** (The KL divergence and negative entropy). The Kullback–Leibler (KL) divergence and the negative entropy are defined for $\mu \ll \bar{\mu}$ and $\mu \ll \rho$, writing $\mu(x) = d\mu/d\rho(x)$, respectively as

$$\mathrm{KL}(\mu|\bar{\mu}) = \int_\mathcal{X} \ln\left(d\mu/d\bar{\mu}(x)\right) d\mu(x), \quad \phi_e(\mu) = \int_\mathcal{X} \ln(\mu(x))\mu(x)d\rho(x) = \mathrm{KL}(\mu|\rho), \tag{13}$$

where $\rho$ is some reference finite measure on $\mathcal{X}$. It is straightforward to show that KL can be written as a Bregman divergence of $\phi_e$ if $\mu \ll \bar{\mu} \ll \rho$, i.e. $D_{\phi_e}(\mu|\bar{\mu}) = \mathrm{KL}(\mu|\bar{\mu})$; hence one can choose $\phi = \phi_e$ for the mirror descent scheme (8). Assumption $(\mathbf{A'}_1)$, guaranteeing that the iterates (8) are well-posed, is satisfied for instance when $\mathcal{M}(\mathcal{X}) = L^1(\mathcal{X})$ and if there exists $\kappa_0, \kappa_1 > 0$ such that $\kappa_0 \leq d\mu_n/d\bar{\mu}(x) \leq \kappa_1$ almost everywhere over $\mathcal{X}$ for any $n$, which is the case for Sinkhorn's and EM iterates. This implies that the first variations of the negative entropy or KL belong to $L^\infty$ at $\mu_n$, see also Remark 5 in Appendix for more details. Moreover, by Lemma 2, exponentiating the dual iteration (9) recovers the classical multiplicative scheme: $\mu_{n+1} = \mu_n e^{-\frac{1}{L}\nabla\mathcal{F}(\mu_n)}$, $n \geq 0$. This observation generalizes to the Iterative Proportional Fitting Procedure, also known as Sinkhorn's algorithm.

KL is a strong Bregman divergence, in the sense that it dominates a wide range of objective functionals $\mathcal{F}$. Indeed, we already know from Section 2 and the idempotence property that $\mathcal{F} = \mathrm{KL}(\cdot|\bar{\mu})$ is 1-relatively smooth w.r.t. $\phi_e$, since for any $\mu, \nu \in \mathrm{dom}(\mathcal{F})$, $D_\mathcal{F}(\nu|\mu) = D_{\phi_e}(\nu|\mu) = \mathrm{KL}(\nu|\mu)$. In Section 4 we will extensively use this fact (sometimes applying KL to joint distributions rather than marginals, e.g. for Sinkhorn's algorithm). This is of crucial importance because $\mathcal{F} = \mathrm{KL}(\cdot|\bar{\mu})$ is not a smooth objective in the "standard" sense (see Example 1) - hence convergence proofs requiring the latter cannot apply -, but it is relatively smooth w.r.t. itself. Indeed the KL diverges for Dirac masses, so is unbounded over the bounded set $\mathcal{P}(\mathcal{X})$, and thus KL does not have subquadratic growth (7) w.r.t. any norm on measures. Other objective functionals can be dominated by KL, such as the Maximum Mean Discrepancy (MMD) for bounded kernels, see Proposition 14 in Appendix D.

## 4 Applications to Sinkhorn and EM with convergence rates

We now analyze the convergence of two algorithms, Sinkhorn and Expectation-Maximization (EM), by showing that they can be written as mirror descent schemes based on the KL divergence, in order to apply the results of Section 3. Note that the relative smoothness was first introduced by Birnbaum et al. (2011) for this very purpose, to study the convergence of Proportional Response Dynamics.

In both the Sinkhorn and EM settings we will be given two probability spaces $(\mathcal{X}, \bar{\mu})$ and $(\mathcal{Y}, \bar{\nu})$. We recall that $\mathcal{P}(\mathcal{X} \times \mathcal{Y})$ denotes the subset of Radon measures $\mathcal{M}_r(\mathcal{X} \times \mathcal{Y})$ with mass 1. We equip $\mathcal{M}_r$ with $L^\infty(\mathcal{X} \times \mathcal{Y})$ as dual space. A joint measure $\pi \in \mathcal{P}(\mathcal{X} \times \mathcal{Y})$ is also called a *coupling* between its first $p_\mathcal{X}\pi$ and second $p_\mathcal{Y}\pi$ marginals. We denote by $\Pi(\bar{\mu}, *)$ the set of couplings having first marginal $\bar{\mu}$ and $\Pi(*, \bar{\nu})$ the set of couplings having second marginal $\bar{\nu}$, and $\Pi(\bar{\mu}, \bar{\nu}) = \Pi(\bar{\mu}, *) \cap \Pi(*, \bar{\nu})$ the couplings with marginals $(\bar{\mu}, \bar{\nu})$. We now recall an instrumental disintegration formula:

Let $\pi, \bar{\pi} \in \mathcal{P}(\mathcal{X} \times \mathcal{Y})$ with $\pi \ll \bar{\pi}$, $K_{\bar{\pi}}(x, dy) = {\bar{\pi}(dx,dy)}/{p_\mathcal{X}\bar{\pi}(dx)}$. We have $\bar{\pi} = p_\mathcal{X}\bar{\pi} \otimes K_{\bar{\pi}}$ and[4]

$$\mathrm{KL}(\pi|\bar{\pi}) = \mathrm{KL}(p_\mathcal{X}\pi|p_\mathcal{X}\bar{\pi}) + \int_\mathcal{X} \mathrm{KL}(K_\pi|K_{\bar{\pi}})\, dp_\mathcal{X}\pi = \mathrm{KL}(p_\mathcal{X}\pi|p_\mathcal{X}\bar{\pi}) + \mathrm{KL}(\pi|p_\mathcal{X}\pi \otimes K_{\bar{\pi}}). \quad (14)$$

This decomposition is at the heart of the two objective functions $F_\mathrm{S}$ and $F_\mathrm{EM}$ considered below.

## 4.1 Sinkhorn

To describe the entropic optimal transport problem we follow Nutz (2021). Consider a cost function $c \in L^\infty(\mathcal{X} \times \mathcal{Y}, \bar{\mu} \otimes \bar{\nu})$ and a regularization parameter $\epsilon > 0$. The entropic optimal transport problem is the minimization problem

$$\mathrm{OT}_\epsilon(\bar{\mu}, \bar{\nu}) = \min_{\pi \in \Pi(\bar{\mu}, \bar{\nu})} \mathrm{KL}(\pi|e^{-c/\epsilon}\bar{\mu} \otimes \bar{\nu}). \quad (15)$$

By adding a constant to $c$ we can assume without loss of generality that $e^{-c/\epsilon}\bar{\mu} \otimes \bar{\nu}$ has mass 1. Since $c$ is bounded, (15) admits a unique solution $\pi_*$. We use a characterization (Nutz, 2021, Theorem 4.2, Lemma 4.9, Section 6) of the set of cyclically invariant couplings, and define it as follows, as the set of couplings $\pi$ that solve an entropic optimal transport problem for their own marginals,

$$\Pi_c = \{\pi \in \mathcal{P}(\mathcal{X} \times \mathcal{Y}) \mid \mathrm{KL}(\pi|e^{-c/\epsilon}\mu \otimes \nu) = \min_{\tilde{\pi} \in \Pi(\mu,\nu)} \mathrm{KL}(\tilde{\pi}|e^{-c/\epsilon}\mu \otimes \nu), (\mu,\nu) = (p_\mathcal{X}\pi, p_\mathcal{Y}\pi)\}. \quad (16)$$

Moreover when $\pi \in \Pi_c$, there exist $f \in L^\infty(\mathcal{X})$ and $g \in L^\infty(\mathcal{Y})$ such that $\pi = e^{(f+g-c)/\epsilon}\mu \otimes \nu$.

The Sinkhorn algorithm in its primal formulation solves (15) by alternating (entropic) projections on $\Pi(\bar{\mu}, *)$ and $\Pi(*, \bar{\nu})$ (Ruschendorf, 1995), i.e. initializing with $\pi_0 \in \Pi_c$, iterate

$$\pi_{n+\frac{1}{2}} = \operatorname*{argmin}_{\pi \in \Pi(\bar{\mu}, *)} \mathrm{KL}(\pi|\pi_n), \quad (17)$$

$$\pi_{n+1} = \operatorname*{argmin}_{\pi \in \Pi(*, \bar{\nu})} \mathrm{KL}(\pi|\pi_{n+\frac{1}{2}}). \quad (18)$$

Let $\mu_n = p_\mathcal{X}\pi_n$. More explicitly, (17) is a "rescaling of the rows", $\pi_{n+\frac{1}{2}}(dx, dy) = \pi_n(dx, dy)\bar{\mu}(dx)/p_\mathcal{X}\pi_n(dx)$, while (18) is a "rescaling of the columns", $\pi_{n+1}(dx, dy) = \pi_{n+\frac{1}{2}}(dx, dy)\bar{\nu}(dy)/p_\mathcal{Y}\pi_{n+\frac{1}{2}}(dy)$. This can be seen as a consequence of (14), as in (17) the first marginal is fixed, so the optimum of (17) is such that the integral term of (14) vanishes. One can also show recursively that $\pi_n \in \Pi_c$ (see Nutz, 2021, Section 6, Lemma 6.22).

Define the constraint set $C = \Pi(*, \bar{\nu})$, $\mathcal{R} = C \cap \Pi_c$ and the objective function

$$F_\mathrm{S}(\pi) = \mathrm{KL}(p_\mathcal{X}\pi|\bar{\mu}). \quad (19)$$

Connections between mirror descent and Sinkhorn iterations for the entropic regularized optimal transport problem were first discovered in Mishchenko (2019); Mensch and Peyré (2020); Léger (2020). We propose yet another mirror descent interpretation of Sinkhorn in the spirit of Léger (2020), and use the primal formulation (17)–(18) directly instead of introducing dual potentials. This is stated in the following Proposition, whose complete proof can be found in Appendix F.2.

**Proposition 5** (Sinkhorn as mirror descent). The Sinkhorn iterations (17) can be written as a mirror descent with objective $F_\mathrm{S}$ and Bregman divergence KL over the constraint $C = \Pi(*, \bar{\nu})$,

$$\pi_{n+1} = \operatorname*{argmin}_{\pi \in C} \langle \nabla_C F_\mathrm{S}(\pi_n), \pi - \pi_n \rangle + \mathrm{KL}(\pi|\pi_n) \text{ with } \nabla_C F_\mathrm{S}(\pi_n) = \ln(d\mu_n/d\bar{\mu}) \in L^\infty(\mathcal{X} \times \mathcal{Y}).$$

*Sketch of proof.* Let $\mu_n = p_\mathcal{X}\pi_n$ where $\pi_n$ is defined in (17). We have the identity:

$$F_\mathrm{S}(\pi_n) + \langle \nabla_C F_\mathrm{S}(\pi_n), \pi - \pi_n \rangle + \mathrm{KL}(\pi|\pi_n) = \mathrm{KL}(\pi|\bar{\mu} \otimes \pi_n/\mu_n) = \mathrm{KL}(\pi|\pi_{n+\frac{1}{2}}).$$

We conclude by taking the argmin over $\pi \in C$. $\qquad\square$

We first show relative smoothness of $F_\mathrm{S}$ relatively to $\phi_e$, as a consequence of the standard KL data processing inequality, i.e. KL of the marginals is smaller than KL of the plans.

---

[4]The last equality uses: $\mathrm{KL}(\pi|p_\mathcal{X}\pi \otimes K_{\bar{\pi}}) = \int \ln\left(\frac{p_\mathcal{X}\pi \otimes K_\pi}{p_\mathcal{X}\pi \otimes K_{\bar{\pi}}}\right) dp_\mathcal{X}\pi \otimes K_\pi = \int_\mathcal{X} \mathrm{KL}(K_\pi|K_{\bar{\pi}})\, dp_\mathcal{X}\pi$.

**Lemma 6.** The functional $F_S$ is convex and is 1-relatively smooth w.r.t. $\phi_e$ over $\mathcal{P}(\mathcal{X} \times \mathcal{Y})$.

*Proof.* Let $\pi, \tilde{\pi} \in \mathcal{P}(\mathcal{X} \times \mathcal{Y})$ with $p_\mathcal{X}\tilde{\pi} \ll p_\mathcal{X}\pi \ll \bar{\mu}$. Then with straightforward computations, $D_{F_S}(\tilde{\pi}|\pi) = \mathrm{KL}(p_\mathcal{X}\tilde{\pi}|p_\mathcal{X}\pi) \geq 0$, so $F_S$ is convex. Then (14) results in $D_{F_S}(\tilde{\pi}|\pi) \leq \mathrm{KL}(\tilde{\pi}|\pi)$. $\qquad\square$

By considering singular first marginals, it is obvious that there exists no $l > 0$ such that $D_{F_S}(\tilde{\pi}|\pi) \geq l\,\mathrm{KL}(\tilde{\pi}|\pi)$ for all $\tilde{\pi}, \pi \in C$; i.e. that relative strong convexity of $F_S$ relatively to $\phi_e$ does not hold *over all* $C$. However, we show in the next Proposition that this inequality actually holds over $\mathcal{R} = \Pi_c \cap C$. Its complete proof can be found in Appendix F.3.

**Proposition 7.** Let $D_c := \frac{1}{2}\sup_{x,y,x',y'}[c(x,y) + c(x',y') - c(x,y') - c(x',y)] < \infty$. For $\tilde{\pi}, \pi \in \Pi_c \cap C$, we have that
$$\mathrm{KL}(\tilde{\pi}|\pi) \leq (1 + 4e^{3D_c/\epsilon})\,\mathrm{KL}(p_\mathcal{X}\tilde{\pi}|p_\mathcal{X}\pi), \tag{20}$$
in other words $F_S$ is $(1 + 4e^{3D_c/\epsilon})^{-1}$-relatively strongly convex w.r.t. KL over $\Pi_c \cap C$.

*Sketch of proof.* For $\pi, \tilde{\pi} \in \Pi_c \cap C$ with their potentials and marginals $(f, g, \mu, \bar{\nu})$ and $(\tilde{f}, \tilde{g}, \tilde{\mu}, \bar{\nu})$ respectively, setting $\|f\|_{\mathrm{var}} = (\sup_\mathcal{X} f) - (\inf_\mathcal{X} f)$, we can derive the bound
$$\epsilon\,\mathrm{KL}(\tilde{\pi}|\pi) \leq \|\tilde{f} - f\|_{\mathrm{var}}\|\tilde{\mu} - \mu\|_{\mathrm{TV}} + \epsilon\,\mathrm{KL}(\tilde{\mu}|\mu). \tag{21}$$

We can then bound the potentials by the marginals using Proposition 15 in Appendix:
$$\|f - \tilde{f}\|_{\mathrm{var}} + \|g - \tilde{g}\|_{\mathrm{var}} \leq 2\epsilon\,e^{3D_c/\epsilon}\left(\|\mu - \tilde{\mu}\|_{\mathrm{TV}} + \|\bar{\nu} - \bar{\nu}\|_{\mathrm{TV}}\right). \tag{22}$$

Then, chaining (21) and (22) we conclude using Pinsker's inequality. $\qquad\square$

We are now ready to recover convergence rates for Sinkhorn leveraging relative smoothness and strong convexity.

**Proposition 8** (Sinkhorn convergence). For all $n \geq 0$, the Sinkhorn iterates verify, for $\pi_*$ the optimum of (15) and $\mu_*$ its first marginal,
$$\mathrm{KL}(\mu_n|\mu_*) \leq \frac{\mathrm{KL}(\pi_*|\pi_0)}{(1 + 4e^{\frac{3D_c}{\epsilon}})\left(\left(1 + 4e^{-\frac{3D_c}{\epsilon}}\right)^n - 1\right)} \leq \frac{\mathrm{KL}(\pi_*|\pi_0)}{n}. \tag{23}$$

*Proof.* Fix $n \geq 0$, we know that $\pi_n, \pi_* \in \mathcal{R} := C \cap \Pi_c$. We conclude by applying Theorem 4 to $\mathcal{F} = F_S$ and $\phi = \mathrm{KL}(\cdot|\pi_*)$, leveraging the results of Lemma 6 and Proposition 7. $\qquad\square$

The linear convergence of Sinkhorn for bounded costs $c$ has been known since at least Franklin and Lorenz (1989) and has then been derived also in the non-discrete case and in multimarginal settings (see Carlier, 2022, and references therein). To derive this linear rate (the first inequality in (23)), we require relative strong convexity of the objective - hence we also fundamentally rely on the boundedness of the cost, as one can see from the assumptions of Proposition 7. Indeed the proof of Proposition 7 relies on the classical result that the soft $c$-transforms are contractions in the Hilbert metric; a result also at the heart of other proofs for the linear rate (see Franklin and Lorenz, 1989; Chen et al., 2016, for a proof). Regarding the sublinear convergence (the second inequality in (23)), Léger (2020) first obtained sublinear rates for unbounded costs leveraging relative smoothness, using (9) formally and through dual iterations on the potentials. In this Section we assumed $c \in L^\infty(\mathcal{X} \times \mathcal{Y})$, in order to manipulate finite quantities in our computations (e.g. first variations in the proof of Proposition 5) - hence the latter can be seen as a convenient working hypothesis. In this paper we derive the same rate as Léger (2020) rigorously with a more direct proof using primal iterations, and complete the picture by recovering linear rates of convergence.

## 4.2 Expectation-Maximization

In this subsection, we show how the EM algorithm can always formally be written as a mirror descent scheme, and, when optimizing the latent variable distribution, results in a convex problem with sublinear convergence rates. Consider the following probabilistic model: we have a latent, hidden random variable $X \in (\mathcal{X}, \bar{\mu})$, an observed variable $Y \in \mathcal{Y}$ distributed as $\bar{\nu}$, and we posit a joint

distribution $p_q(dx, dy)$ parametrized by an element $q$ of some given set $\mathcal{Q}$. As presented in Neal and Hinton (1998), the goal is to infer $q$ by solving

$$\min_{q \in \mathcal{Q}} \mathrm{KL}(\bar{\nu}|p_{\mathcal{Y}} p_q), \tag{24}$$

where $p_{\mathcal{Y}} p_q(dy) = \int_{\mathcal{X}} p_q(dx, dy)$. The EM approach starts by minimizing a surrogate function of $q$ upperbounding $\mathrm{KL}(\bar{\nu}|p_{\mathcal{Y}} p_q)$. For any $\pi \in \Pi(*, \bar{\nu})$, by the data processing inequality,

$$\mathrm{KL}(\bar{\nu}|p_{\mathcal{Y}} p_q) \leq \mathrm{KL}(\pi|p_q) =: L(\pi, q).$$

Again, as a consequence of the disintegration formula (14), there is equality if and only if

$$\pi(dx, dy) = p_q(dx, dy)\bar{\nu}(dy)/p_{\mathcal{Y}} p_q(dy). \tag{25}$$

EM then proceeds by alternate minimizations of $L(\pi, q)$ (see Neal and Hinton, 1998, Theorem 1):

$$q_n = \operatorname*{argmin}_{q \in \mathcal{Q}} \mathrm{KL}(\pi_n|p_q), \tag{26}$$

$$\pi_{n+1} = \operatorname*{argmin}_{\pi \in \Pi(*, \bar{\nu})} \mathrm{KL}(\pi|p_{q_n}). \tag{27}$$

The above formulation consists in (26), optimizing the parameters $q_n$ at step $n$ (M-step), and then (27), optimizing the joint distribution $\pi_{n+1}$ at step $n + 1$ (E-step). We choose this order to highlight the analogy of (27) with Sinkhorn's (18). The minimization (27) corresponds to taking an explicit expectation, according to (25) justifying the denomination. On the contrary making explicit the M-step is often difficult.

Define the constraint set $C = \Pi(*, \bar{\nu})$ and the, possibly non-convex, objective function

$$F_{\mathrm{EM}}(\pi) = \inf_{q \in \mathcal{Q}} \mathrm{KL}(\pi|p_q). \tag{28}$$

We first show that EM can be formally written as a mirror descent scheme in the following Proposition, whose precise statement with additional assumptions can be found in Appendix F.4.

**Proposition 9** (EM as mirror descent, formal)**.** The EM iterations (26)–(27) can be written as a mirror descent iteration with objective function $F_{\mathrm{EM}}$, Bregman potential $\phi_e$ and constraints $C$,

$$\pi_{n+1} = \operatorname*{argmin}_{\pi \in C} \langle \nabla_C F_{\mathrm{EM}}(\pi_n), \pi - \pi_n \rangle + \mathrm{KL}(\pi|\pi_n) \text{ with } \nabla_C F_{\mathrm{EM}}(\pi_n) = \ln(d\pi_n/dp_{q_n}). \tag{29}$$

*Sketch of proof.* Let $\pi_n$ be the current EM iterate. Formally, we use the envelope theorem to differentiate $F_{\mathrm{EM}}$ and find that $\nabla_C F_{\mathrm{EM}}(\pi_n) = \ln(d\pi_n/dp_{q_n})$ (see Appendix F.4 for a justification based on directional derivatives and Milgrom and Segal (2002, Theorem 3)). Then for any coupling $\pi$, we have the identity

$$F_{\mathrm{EM}}(\pi_n) + \langle \nabla_C F_{\mathrm{EM}}(\pi_n), \pi - \pi_n \rangle + \mathrm{KL}(\pi|\pi_n) = \mathrm{KL}(\pi|p_{q_n}).$$

Thus (29) matches (27). $\qquad\square$

Since $F_{\mathrm{EM}}$ is in general non-convex, we cannot apply outright the framework developed in Section 3. There is however one direct case of $p_q$ making $F_{\mathrm{EM}}$ convex, by optimizing only over its first marginal.

**Latent EM.** We consider the case where $p_q(dx, dy) = \mu(dx)K(x, dy)$, i.e. $p_q$ is of the form $\mu \otimes K$, with $\mathcal{Q} = \mathcal{P}(\mathcal{X})$ and $K$ kept fixed along iterations. In other words, we choose to only optimize over the density of the latent variable, and keep the mixture parameters fixed. Here $K$ can be interpreted as the conditional distribution of $Y$ given $X$, $K(x, dy) = \mathbb{P}(Y = y \mid X = x)$. We consider in the following Gibbs distributions with $K(x, dy) = e^{-c(x,y)}\bar{\nu}(dy)$ with $c$ uniformly bounded. The term $c$ can be interpreted as a cost similarly to the entropic optimal transport (15).

**Remark 3** (Various EM)**.** The general goal of EM is to fit, through the objective function $F_{\mathrm{EM}}$, a parametric distribution, e.g. a mixture of Gaussians, to some observed data $Y$. One needs to estimate both the latent variable distribution on $X$ (i.e. weights of each Gaussian) and the parameters of conditionals $P(Y|X = x)$ (e.g. means and covariances of each Gaussian). Latent EM focuses on learning the mixture weights, since it consists in optimizing over the nonparametric latent distribution $\mu$, which can be continuous or discrete. In contrast, the parametric setting considered by Kunstner et al. (2021), who obtained $\mathcal{O}(1/n)$ rates of convergence in KL for EM, can be seen as complementary to ours since they consider a fixed $\mu$ and a variable exponential mixture $K_\theta$, with $q = \theta$.

Parametrizing by the first marginal, EM iterations (26)–(27) takes the following form for Latent EM:

$$\mu_n = \underset{\mu \in \mathcal{P}(\mathcal{X})}{\operatorname{argmin}} \mathrm{KL}(\pi_n | \mu \otimes K), \qquad (30)$$

$$\pi_{n+1} = \underset{\pi \in \Pi(*, \bar{\nu})}{\operatorname{argmin}} \mathrm{KL}(\pi | \mu_n \otimes K). \qquad (31)$$

First, we necessarily have from (30) that $\mu_n = p_{\mathcal{X}} \pi_n$. Indeed, from the disintegration formula (14), (30) corresponds to minimizing over first marginals. Then, since the E-step (31) corresponds to computing (25), we can rewrite (30)–(31) as:

$$\mu_{n+1}(\cdot) = \int_{\mathcal{Y}} \pi_{n+1}(\cdot, dy) = \mu_n(\cdot) \int_{\mathcal{Y}} \frac{K(\cdot, dy)\bar{\nu}(dy)}{\int_{\mathcal{X}} K(x, dy)\mu_n(dx)}. \qquad (32)$$

Define $F_{\mathrm{LEM}}(\pi) := \inf_{\mu \in \mathcal{P}(\mathcal{X})} \mathrm{KL}(\pi | \mu \otimes K)$. Notice that by disintegration (14), $F_{\mathrm{LEM}}$ takes the form $F_{\mathrm{LEM}}(\pi) = \mathrm{KL}(\pi | p_{\mathcal{X}} \pi \otimes K)$. To take care of the initialization, we define the operator $T_K : \mu \in \mathcal{P}(\mathcal{X}) \mapsto \int_{\mathcal{X}} \mu(dx)K(x, \cdot) \in \mathcal{M}_r(\mathcal{Y})$, with $K(x, dy) = k(x, y)\bar{\nu}(dy)$ and take $\mu_0 = e^{f_0}\bar{\mu}$ with $f_0 \in L^\infty$ and assume that $T_K \bar{\mu} \gg \bar{\nu}$ (in other words we assume that the mixture applied to the latent space is compatible with all the observations).

**Proposition 10** (Latent EM as mirror descent). The latent EM iterations (30)–(31) can be written as mirror descent with objective $F_{\mathrm{LEM}}$, Bregman potential $\phi_e$ and the constraints $C = \Pi(*, \bar{\nu})$,

$$\pi_{n+1} = \underset{\pi \in C}{\operatorname{argmin}} \langle \nabla_C F_{\mathrm{LEM}}(\pi_n), \pi - \pi_n \rangle + \mathrm{KL}(\pi | \pi_n) \text{ with } \nabla_C F_{\mathrm{LEM}}(\pi_n) = \ln\left(\frac{d\pi_n}{d(\mu_n \otimes K)}\right) \in L^\infty.$$

*Proof.* Similarly to Proposition 9, we have the identity

$$F_{\mathrm{LEM}}(\pi_n) + \langle \nabla_C F_{\mathrm{LEM}}(\pi_n), \pi - \pi_n \rangle + \mathrm{KL}(\pi | \pi_n) = \mathrm{KL}(\pi | p_{\mathcal{X}} \pi_n \otimes K),$$

where $\nabla_C F_{\mathrm{LEM}}(\pi_n) = \ln(\pi_n / p_{\mathcal{X}} \pi_n \otimes K) \in L^\infty(\mathcal{X}, \mathbb{R})$, see Appendix F.5 for rigorous justifications. Since $\mu_n = p_{\mathcal{X}} \pi_n$ due to (30), we conclude by minimizing over $\pi \in C$. $\qquad \square$

We are now ready to state convergence rates of latent EM in the following proposition. The reader may refer to Appendix F.6 for a complete proof.

**Proposition 11** (Convergence rate for Latent EM). Set $\mu_* \in \operatorname{argmin}_{\mu \in \mathcal{P}(\mathcal{X})} \mathrm{KL}(\bar{\nu} | T_K(\mu))$. The functional $F_{\mathrm{LEM}}$ is convex and 1-smooth relative to $\phi_e$. Moreover for $\pi_0 \in \Pi(*, \bar{\nu})$,

$$\mathrm{KL}(\bar{\nu} | T_K \mu_n) \le \mathrm{KL}(\bar{\nu} | T_K \mu_*) + \frac{\mathrm{KL}(\mu_* | \mu_0) + \mathrm{KL}(\bar{\nu} | T_K \mu_*) - \mathrm{KL}(\bar{\nu} | T_K \mu_0)}{n}.$$

*Sketch of proof.* By the disintegration formula (14), we can decompose $F_{\mathrm{LEM}}$ as

$$F_{\mathrm{LEM}}(\pi) = \mathrm{KL}(\bar{\nu} | p_{\mathcal{Y}}(p_{\mathcal{X}} \pi \otimes K)) + \int \mathrm{KL}(\pi / \bar{\nu} | (p_{\mathcal{X}} \pi \otimes K) / p_{\mathcal{Y}}(p_{\mathcal{X}} \pi \otimes K)) d\bar{\nu}, \qquad (33)$$

and show then straightforwardly that $\pi_*$ defined by $\pi_*(dx, dy) = \mu_*(dx)k(x, dy)\bar{\nu}(dy)/(T_K \mu_*)(dy)$ is a minimizer of $F_{\mathrm{LEM}}$ with $F_{\mathrm{LEM}}(\pi^*) = \mathrm{KL}(\bar{\nu} | T_K \mu_*)$. Then, by the disintegration formula (14) and linearity of the Bregman divergence, $\mathrm{KL}(\pi | \tilde{\pi}) = D_{F_{\mathrm{S}}}(\pi | \tilde{\pi}) + D_{F_{\mathrm{LEM}}}(\pi | \tilde{\pi})$, hence $F_{\mathrm{LEM}}$ is 1-relatively smooth w.r.t. $\phi_e$. Consequently, Theorem 4 yields:

$$F_{\mathrm{LEM}}(\pi_n) \le F_{\mathrm{LEM}}(\pi_*) + \frac{\mathrm{KL}(\pi_* | \pi_0)}{n},$$

Finally: $\mathrm{KL}(\bar{\nu} | T_K \mu_n) = \mathrm{KL}(p_{\mathcal{Y}} \pi_n | p_{\mathcal{Y}}(p_{\mathcal{X}} \pi_n \otimes K)) \le \mathrm{KL}(\pi_n | p_{\mathcal{X}} \pi_n \otimes K) = F_{\mathrm{LEM}}(\pi_n)$. $\qquad \square$

**Remark 4** (Richardson–Lucy). Interestingly, the iterations (32) of latent EM correspond precisely to that of Richardson–Lucy deconvolution (Richardson, 1972; Lucy, 1974) where $K$ is a known convolution and one aims at recovering the original signal $\mu_*$ based on the observations $\bar{\nu}$. Thus our proof yields rates of convergence for this other algorithm from signal processing, a novel result to the best of our knowledge.

**Conclusion:** We have provided a rigorous proof of convergence of mirror descent under relative smoothness and convexity, which holds in the infinite-dimensional setting of optimization over measure spaces. The latter condition can handle objective functionals that are not smooth in the standard sense, such as the ubiquitous KL. It enabled us to provide a new and simple way to derive rates of convergence for Sinkhorn's algorithm. We also derived new convergence rates for EM when restricted to the latent distribution, obtaining complementary rates to Kunstner et al. (2021).

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
