# Appendix

## A  Definition of Gâteaux and Fréchet derivatives

We first recall the notion of Gâteaux and Fréchet derivatives for $\mathcal{F} : \mathcal{M}(\mathcal{X}) \to \mathbb{R} \cup \{\pm\infty\}$ where $\mathcal{M}(\mathcal{X})$ is a topological vector space (Aliprantis and Border, 2006, Chapter 7, pp.267,273), see also Phelps (1989, Section 1) for Banach spaces.

**Definition 5.** The function $\mathcal{F}$ is said to be Gâteaux differentiable at $\nu$ if there exists a linear operator $\nabla F(\nu) : \mathcal{M}(\mathcal{X}) \to \mathbb{R}$ such that for any direction $\mu \in \mathcal{M}(\mathcal{X})$:

$$\nabla \mathcal{F}(\nu)(\mu) = \lim_{h \to 0} \frac{\mathcal{F}(\nu + h\mu) - \mathcal{F}(\nu)}{h}. \tag{34}$$

The operator $\nabla \mathcal{F}(\nu)$ is called the Gâteaux derivative of $\mathcal{F}$ at $\nu$, and if it exists, it is unique.

**Definition 6.** If $\mathcal{M}(\mathcal{X})$ is a normed space, the function $\mathcal{F}$ is said to be Fréchet differentiable at $\nu$ if there exists a bounded linear form $\delta \mathcal{F}(\nu, \cdot) : \mathcal{M}(\mathcal{X}) \to \mathbb{R}$ such that

$$\mathcal{F}(\nu + h\mu) = \mathcal{F}(\mu) + h\delta \mathcal{F}(\nu, \mu) + ho(\|\mu\|_{\mathcal{M}(\mathcal{X})}) \tag{35}$$

Equivalently, the operator $\delta \mathcal{F}(\nu, \cdot)$ is called the Fréchet derivative of $\mathcal{F}$ at $\nu$ if it is a Gâteaux derivative of $\mathcal{F}$ at $\nu$ and the limit (34) holds uniformly in $\mu$ in the unit ball (or unit sphere) in $\mathcal{M}(\mathcal{X})$.

If $\mathcal{F}$ is Fréchet differentiable, then it is also Gâteaux differentiable, and its Fréchet and Gâteaux derivatives agree: $\nabla \mathcal{F}(\nu)(\mu) = \delta \mathcal{F}(\nu, \mu)$.

## B  Additional details on the well-posedness of the mirror descent scheme

Recall Assumption (**A'**$_1$)(Lower semicontinuity and coercivity): (i) the set $C$ is $\tau$-closed in $\mathcal{M}(\mathcal{X})$, the functionals $\mathcal{G}_n(\cdot) := d^+\mathcal{F}(\mu_n)(\cdot - \mu_n)$ and $D_\phi(\cdot|\mu_n)$ are proper and $\tau$-l.s.c. when restricted to $C$, and the functional $\mathcal{G}_n + D_\phi(\cdot|\mu_n) + i_C{}^5$ has at least one $\tau$-compact sublevel set. (ii) For each $n \geq 0$, the first variations $\nabla_C \phi(\mu_n)$ exist. (iii) The iterates belong to $\mathcal{R}$.

**Case where $\mathcal{F}$ has first variations.** Equip $\mathcal{M}(\mathcal{X})$ with a topology $\tau$ that is stronger than the $\mathcal{M}(\mathcal{X})^*$-weak topology. If $\mathcal{F}$ has first variations, then we can even remove $\mathcal{G}_n(\cdot)$ from Assumption (**A'**$_1$), since $\mathcal{G}_n(\cdot)$ is linear on the set of interest and $\tau$-l.s.c. Whence we get the simpler assumption

  (**A''**$_1$)  (Lower semicontinuity and coercivity) For each $n \geq 0$, the iterates belong to $\mathcal{R}$ and the first variations $\nabla_C \phi(\mu_n)$ and $\nabla_C \mathcal{F}(\mu_n)$ exist. Moreover the set $C$ is $\tau$-closed in $\mathcal{M}(\mathcal{X})$, the functional $\phi$ is proper and $\tau$-l.s.c. when restricted to $C$, and $\phi(\cdot)$ has at least one $\tau$-compact sublevel set when restricted to $C \cap \mathrm{dom}(\mathcal{F})$.

**Weakly compact sets of $\mathcal{M}(\mathcal{X})$.** In finite dimensions, a set is compact iff bounded and closed; however, in infinite dimensions, characterizing compact sets is more delicate. Below we recall some classical set of conditions that guarantee (weak) compactness or lower semicontinuity.

If $\mathcal{M}(\mathcal{X})$ is a reflexive Banach space, then the weakly compact sets are just the bounded weakly closed sets, as a consequence of the Banach–Alaoglu theorem (see e.g. Attouch et al., 2014, Theorem 2.4.2). In other cases, one needs more specific theorems such as Dunford–Pettis' theorem for $L^1(\rho)$ (see e.g. Attouch et al., 2014, Theorem 2.4.5). Since we are dealing with convex functions, for normed $\mathcal{M}(\mathcal{X})$, strongly closed sublevel sets are also weakly closed, a result known as Mazur's lemma. So the notions of weakly l.s.c. and strongly l.s.c. convex functions coincide, as recalled in Attouch et al. (2014, Theorem 3.3.3).

We now regroup some known properties of KL, in particular to show that Assumption (**A'**$_1$) holds for $\phi = \phi_e$.

**Remark 5** (Properties of KL). For compact $\mathcal{X}$, the domain of the negative entropy $\phi_e$ is strictly included in $L^1_+(\mathcal{X})$, contains $L^q_+(\mathcal{X})$ for $q > 1$, and is of empty interior for the norm/strong topology of $L^q(\mathcal{X})$ (Resmerita, 2005, Lemma 4.1). Regarding the use of $\phi = \phi_e$ in (8), one can for instance

---

$^5 i_C$ denotes the indicator function of the set $C$, defined by $i_C(\mu) = 0$ if $\mu \in C$, $+\infty$ otherwise for any $\mu \in \mathcal{M}(\mathcal{X})$. Notice that $i_C$ being $\tau$-l.s.c. is equivalent to $C$ being $\tau$-closed in $\mathcal{M}(\mathcal{X})$.

take $\mathcal{M}(\mathcal{X}) = L^1(\mathcal{X})$ equipped with the weak topology induced by $L^\infty(\mathcal{X})$ and the Lebesgue measure as reference. We have that $\mathcal{P}(X) \cap L^1(\mathcal{X})$ is weakly closed and that KL and $\phi_e$ are strictly convex, weakly l.s.c. and have weakly compact sublevel sets in $L^1(\mathcal{X})$ by Eggermont (1993, Lemma 2.1, 2.3) (see also Resmerita and Anderssen, 2007, Section 3). By Resmerita (2005, Lemma 4.1), a sufficient condition for KL (resp. $\phi_e$) to have a first variation in $L^\infty$ at $\mu$ is that there exists $\kappa_0, \kappa_1 > 0$ such that $\kappa_0 \le {}^{d\mu}/_{d\bar{\mu}}(x) \le \kappa_1$ almost everywhere over $\mathcal{X}$ (resp. $\phi_e$ for $\bar{\mu} = \rho$). KL is not Gâteaux-differentiable for non-finite $\mathcal{X}$ as recalled for instance in Butnariu and Resmerita (2006, p12) and Santambrogio (Remark 7.13 2015, p239).

As a follow-up of Remark 2, we now give some known conditions for a sum of subdifferentials to be the subdifferential of the sum.

**Remark 6** (About the proof of convergence in Theorem 4). Our proof of Theorem 4 resembles the one of Lu et al. (2018), which also relies on a three-point inequality as stated in Lemma 3. However, the proof of the latter inequality in finite dimensions uses a formula of the form $\partial(\mathcal{G} + D_\phi) = \partial\mathcal{G} + \partial D_\phi$ as in Chen and Teboulle (1993, Lemma 3.2), but which is harder to derive in infinite dimensions. Such an equality between subdifferentials can be obtained typically under at least three (non-equivalent) conditions for convex and l.s.c. $\mathcal{G}$ and $\phi$ over a Banach space $\mathcal{M}(\mathcal{X})$: (i) having $\cup_{\lambda \ge 0} \lambda(\text{dom}(\mathcal{G}) - \text{dom}(D_\phi))$ to be a closed vector space of $\mathcal{M}(\mathcal{X})$ (Attouch and Brezis, 1986); (ii) having a non-empty (quasi) relative interior of $(\text{dom}(\mathcal{G}) - \text{dom}(D_\phi))$ (Borwein and Goebel, 2003); (iii) continuity of $D_\phi$ or $\mathcal{G}$ at least at some $\mu \in \text{dom}(\mathcal{G}) \cap \text{dom}(D_\phi)$ (Peypouquet, 2015, Theorem 3.30). Condition (iii) does not hold when $\mathcal{F}$ and $D_\phi$ are chosen as the KL divergence (defined below in Example 2) in none of the spaces we consider since KL is not continuous, its domain being of empty interior. The other conditions are difficult to verify for given functionals. For instance, $\text{dom}(\phi)$ is not explicit for the negative entropy (see Example 2). On the contrary, by favoring directional derivatives and first variations, we circumvent most of the difficulties related to (sub)differentiability.

## C   Additional technical results

**Lemma 12.** Let $f$ be a proper function over a vector space $\mathcal{Y}$ with values in $\mathbb{R} \cup \{+\infty\}$. The following conditions are equivalent:

   i) $f$ is convex;

   ii) $\text{dom}(f)$ is convex, and, for all $x, y \in \text{dom}(f)$, $d^+f(x)(y - x)$ exists, with value in $\mathbb{R} \cup \{-\infty\}$, and we have $f(x) + d^+f(x)(y - x) \le f(y)$, i.e. $D_f(y|x) \ge 0$;

   iii) $\text{dom}(f)$ is convex, and, for all $x, y \in \text{dom}(f)$, $d^+f(x)(y - x)$ exists, with value in $\mathbb{R} \cup \{-\infty\}$, and we have $d^+f(x)(y - x) + d^+f(y)(x - y) \le 0$.

The lemma immediately extends to strictly convex functions by taking strict inequalities.

*Proof.* Given $x, y \in \text{dom}(f)$, define for any $\lambda \in [0, 1]$ $u_\lambda = x + \lambda(y - x)$. Assuming (i), then

$$f(u_\lambda) \le \lambda f(y) + (1 - \lambda)f(x)$$

$$f(x) + \frac{f(u_\lambda) - f(x)}{\lambda} \le f(y),$$

which yields (ii) by the decreasingness of differential quotients discussed in Remark 1.

Assuming (ii), we just sum the two inequalities ($f(x) + d^+f(x)(y - x) \le f(y)$) and ($f(x) + d^+f(x)(y - x) \le f(y)$), to derive (iii).

The last implication to show is (iii)$\Rightarrow$ (i) which requires to perform an integration. Assume that (iii) holds, we want to show that $f(u_\lambda) \le \lambda f(y) + (1 - \lambda)f(x)$. Set $g(\lambda) := f(u_\lambda)$ and denote by $g'_+(\lambda)$ (resp. $g'_-(\lambda)$) its right (resp. left) derivative, both derivatives exist with value in $\mathbb{R} \cup \{-\infty\}$ for any $\lambda \in (0, 1)$ since

$$d^+f(u_\lambda)(y - x) = \lim_{h \to 0^+} \frac{f(u_\lambda + h(y - x)) - f(x)}{h} = g'_+(\lambda)$$

similarly $d^+f(u_\lambda)(x - y) = -g'_-(\lambda)$. Consequently, for all $0 < \lambda_1 < \lambda_2 < 1$, applying (iii) to $u_{\lambda_1}$ and $u_{\lambda_2}$, we have that $g'_+(\lambda_1) \le g'_-(\lambda_2)$. We now show that $\lambda \mapsto g'_-(\lambda)$ is increasing over $(0, 1)$.

We just have to show that $g'_-(\lambda_1) \le \sup_{\lambda \in (0,\lambda_2)} g'_+(\lambda)$. By contradiction, we could fix $\lambda_1 \in (0, \lambda_2)$ and $\epsilon > 0$ such that, for all $\lambda \in (0, \lambda_2)$, $g'_-(\lambda_1) \ge g'_+(\lambda) + \epsilon$. By definition of the directional derivatives, we can then fix $\delta_1 \in (0, \lambda_1)$ and $\lambda \in (\lambda_1 - \delta_1, \lambda_1)$ such that for all $h_0 \in (0, \delta_1)$

$$|g'_-(\lambda_1) + \frac{g(\lambda_1 - h_0) - g(\lambda_1)}{h_0}| \le \epsilon/4 \quad ; \quad |g'_+(\lambda) - \frac{g(\lambda_1) - g(\lambda)}{\lambda_1 - \lambda}| \le \epsilon/4$$

whence

$$\frac{g(\lambda_1) - g(\lambda_1 - h_0)}{h_0} \ge \frac{g(\lambda_1) - g(\lambda)}{\lambda_1 - \lambda} + \epsilon/2$$

which leads to a contradiction for $h_0 = \lambda_1 - \lambda$. Therefore $\lambda \to g'_-(\lambda)$ is increasing over $(0, 1)$, upper bounded by $g'_-(y) = d^+f(u_1)(x - y)$. Since $g$ has both left and right derivatives, it is continuous over $[0, 1]$. We can now apply (iii) to $x$ and $u_\lambda$, use the positive homogeneity of the directional derivative (which always holds by definition), and integrate over $(0, 1)$ since the function $g'_-$ is Riemann-integrable,

$$0 \ge d^+f(x)(u_\lambda - x) + d^+f(u_\lambda)(x - u_\lambda)$$
$$= \lambda d^+f(x)(y - x) + \lambda d^+f(u_\lambda)(x - y)$$
$$0 \ge d^+f(x)(y - x) - \int_0^1 g'_-(\lambda)d\lambda$$
$$= d^+f(x)(y - x) + g(0) - g(1)$$
$$= d^+f(x)(y - x) + f(x) - f(y),$$

which concludes the proof. $\qquad\square$

Below, we derive some useful characterizations of relative smoothness, by analogy with Bauschke et al. (2017, Proposition 1) for differentiable functions in finite dimensions. Similar results hold for relative convexity by the same arguments.

**Lemma 13.** The following conditions are equivalent:

(i) $\mathcal{F}$ is $L$-smooth relative to $\phi$ over $C$;

(ii) $L\phi - \mathcal{F}$ is convex on $C \cap \text{dom}(\phi) \cap \text{dom}(\mathcal{F})$;

and, if the first variations of $\mathcal{F}$ and $\phi$ over $C$ evaluated at $\mu, \nu \in C \cap \text{dom}(\phi) \cap \text{dom}(\mathcal{F})$ exist,

(iii) $\langle \nabla_C \mathcal{F}(\mu) - \nabla_C \mathcal{F}(\nu), \mu - \nu \rangle \le L \langle \nabla_C \phi(\mu) - \nabla_C \phi(\nu), \mu - \nu \rangle$

*Proof.* This is a consequence of Lemma 12 applied to $\psi(\mu) = L\phi(\mu) - \mathcal{F}(\mu)$. More precisely, condition (i) can be written as $d^+\psi(\mu)(\nu - \mu) \le \psi(\nu) - \psi(\mu)$ which is equivalent to the convexity of $\psi$ by Lemma 12, hence (i)⇔(ii). Provided the first variations of $\mathcal{F}$ and $\phi$ over $C$ exist, assuming (i) and (iii) boils down to Lemma 12-iii). Conversely, assuming (iii), we use Lemma 12-iii) and the linearity of the first variation (3). $\qquad\square$

# D  Smoothness of the Maximum Mean Discrepancy relatively to the KL divergence

Let $k : \mathcal{X} \times \mathcal{X} \to \mathbb{R}$ be a positive semi-definite kernel, $\mathcal{H}_k$ its corresponding Reproducing Kernel Hilbert Space (Steinwart and Christmann, 2008). The space $\mathcal{H}_k$ is a Hilbert space with inner product and norm $\| \cdot \|_{\mathcal{H}_k}$ satisfiying the reproducing property: for all $f \in \mathcal{H}_k$ and $x \in \mathcal{X}$, $f(x) = \langle f, k(x, \cdot) \rangle_{\mathcal{H}_k}$. For any $\mu \in \mathcal{P}(\mathcal{X})$ such that $\int \sqrt{k(x,x)}d\mu(x) < \infty$, the kernel mean embedding of $\mu$, $m_\mu = \int k(x, \cdot)d\mu(x)$, is well-defined, belongs to $\mathcal{H}_k$, and $\mathbb{E}_\mu[f(X)] = \langle f, m_\mu \rangle_{\mathcal{H}_k}$ (Smola et al., 2007). The kernel $k$ is said to be characteristic when such mean embedding is injective, that is, when any probability distribution is associated to a unique mean embedding. In this case, the kernel defines a distance between probability distributions referred to as the Maximum Mean Discrepancy (MMD), defined through the square norm of the difference between mean embeddings:

$$\text{MMD}^2(\mu, \bar\mu) = \|m_\mu - m_{\bar\mu}\|^2_{\mathcal{H}_k}$$

Interestingly, as soon as the kernel is bounded, the MMD is relatively smooth with respect to $\phi_e$, see Proposition 14 below. Notice that, thanks to the reproducing property, $\mu \mapsto \mathrm{MMD}(\mu, \bar{\mu})$ is strictly convex whenever the kernel $k$ is characteristic, as it is the case for the Gaussian kernel. Similarly to KL, the MMD can be written as a Bregman divergence of $\phi_k(\mu) = \|m_\mu\|^2_{\mathcal{H}_k} = \int k(x, x')d\mu(x)d\mu(x')$.

**Proposition 14.** Let $\phi_e : \mu \mapsto \int \log(\mu)d\mu$ and fix $\nu \in \mathcal{P}(\mathcal{X})$. Take $k : \mathcal{X} \times \mathcal{X} \to \mathbb{R}$ to be a bounded semipositive definite kernel, i.e. $c_k = \sup_{x \in \mathcal{X}} k(x, x) < \infty$. The squared Maximum Mean Discrepancy $\mathrm{MMD}^2(\cdot, \nu)$ is $4c_k$-smooth relative to $\phi_e$.

*Proof.* Let $\mu, \nu \in \mathcal{P}(\mathcal{X})$ and $f_{\mu, \bar{\mu}} = \int k(x, \cdot)d\mu(x) - \int k(x, \cdot)d\bar{\mu}(x) = \frac{1}{2}\nabla \mathrm{MMD}^2(\mu, \bar{\mu})$. We have:

$$
\begin{aligned}
\langle \nabla \mathrm{MMD}^2(\mu, \bar{\mu}) - \nabla \mathrm{MMD}^2(\nu, \bar{\mu}), \mu - \nu \rangle &\leq \|\nabla \mathrm{MMD}^2(\mu, \bar{\mu}) - \nabla \mathrm{MMD}^2(\nu, \bar{\mu})\|_\infty \|\mu - \nu\|_{TV} \\
&\leq 2\|f_{\mu, \bar{\mu}} - f_{\nu, \bar{\mu}}\|_\infty \|\mu - \nu\|_{TV} \\
&\leq 2 \sup_{y \in \mathcal{X}} |\int k(x, y)d\mu(x) - \int k(x, y)d\nu(x)| \|\mu - \nu\|_{TV}
\end{aligned}
$$

since by the reproducing property and Cauchy-Schwarz inequality, $k(x, y) = \langle k(x, \cdot), k(y, \cdot) \rangle \leq \|k(x, \cdot)\|_k \|k(y, \cdot)\|_k = \sqrt{k(x, x)k(y, y)} \leq c_k$, and $y \mapsto k(x, y)$ is measurable,

$$
\begin{aligned}
&\leq 2c_k \sup_{\substack{f : \mathcal{X} \to [-1, 1] \\ f \text{measurable}}} |\int f(x)d\mu(x) - \int f(x)d\nu(x)| \|\mu - \nu\|_{TV} \\
&\leq 2c_k \|\mu - \nu\|^2_{TV} \leq 4c_k(\mathrm{KL}(\mu|\nu) + \mathrm{KL}(\nu|\mu)) = 4c_k \langle \nabla \phi_e(\mu) - \nabla \phi_e(\nu), \mu - \nu \rangle,
\end{aligned}
$$

where the last inequality results from Pinsker's inequality. We conclude by using Lemma 13. $\qquad \square$

**Remark 7.** (Case of neural network optimization). It is interesting to quantify the constant $c_k$ for some kernels of interest, for instance when optimizing an infinite-width one hidden layer neural network as in Arbel et al. (2019). Consider a regression task where the labelled data $(z, y) \sim P$ where $P$ denotes some fixed data distribution. For any input $z$, the output of a single hidden layer neural network parametrized by $w \in \mathcal{X}$ can be written:

$$
f_w(z) = \frac{1}{N} \sum_{j=1}^{N} a_j \sigma(\langle b_j, z \rangle) = \int_{\mathcal{X}} \phi(z, w)d\mu(w),
$$

where $a_j$ and $b_j$ denote output and input weights of neuron $j = 1, \ldots, N$ respectively, $w_j = (a_j, b_j)$ and $\mu = 1/N \sum_{j=1}^{N} \delta_{w_j}$. In the infinite-width setting, the limiting risk in this regression setting is written for any distribution $\mu \in \mathcal{P}(\mathcal{X})$ on the weights as $\mathbb{E}_{(z,y) \sim P}[\|y - \int \phi(z, w)d\mu(w)\|^2]$. When the model is well-posed, i.e. there exists a distribution $\mu^*$ over weights such that $\mathbb{E}[y|z = \cdot] = \int \phi(\cdot, w)d\mu^*(w)$, then the limiting risk writes as an MMD with $k(w, w') = \mathbb{E}_{z \sim P}[\phi(z, w)^T\phi(z, w)]$ (Arbel et al. (2019, Proposition 20)). Hence, bounding $c_k = \sup_{w \in \mathcal{X}} k(w, w)$ depends on the choice of the activation function $\sigma$ and on bounding the output weights. If $\sigma$ is bounded (e.g. $\sigma$ is the sigmoid activation) then bounding $c_k$ corresponds to bounding the output weights. If $\sigma$ is the RelU activation, then bounding $c_k$ depends on bounding both input and output weights as well, and on the data distribution $P$.

# E  Related work - Optimization over measures using the Wasserstein geometry

In this section, we attempt to clarify the differences between the (Radon) vector space geometry considered in this paper and the Wasserstein geometry, developed in particular in Otto (2001); Villani (2003); Ambrosio et al. (2008).

Given an optimisation problem over $\mathcal{P}(\mathcal{X})$ the set of probability distributions over $\mathcal{X}$, one can consider different geometries over $\mathcal{P}(\mathcal{X})$. The one adopted in our paper casts $\mathcal{P}(\mathcal{X})$ as a subset of a

normed space of measures, such as $L^2(\rho)$ where $\rho$ is a reference measure, or Radon measures. In this space, the shortest distance paths between measures are given by their square-norm distance. Moreover in this setting, one can consider the duality of measures with continuous functions and the mirror descent algorithm, as we do in this work.

In contrast, another possibility is to restrict $\mathcal{P}(\mathcal{X})$ to the probability distributions with bounded second moments, denoted $\mathcal{P}_2(\mathcal{X})$, equipped with Wasserstein-2 ($W_2$) distance. The space $(\mathcal{P}_2(\mathcal{X}), W_2)$, called the Wasserstein space, is a metric space equipped with a rich Riemannian structure (often referred to as "Otto calculus") where the shortest distance paths are given by the $W_2$ distance and associated geodesics. In this setting, one can leverage the Riemannian structure to discretize ($W_2$) gradient flows and consider algorithms such as ($W_2$) gradient descent, in analogy with Riemannian gradient descent.

While both frameworks yield optimisation algorithms on measure spaces, the geometries and algorithms are very different. Both the notion of convexity (along $L^2$ versus $W_2$ geodesics) and of gradients (first variation vs gradient of first variation) differ; and by extension so do many definitions. Consequently, the conditions needed for the convergence of mirror descent and $W_2$ gradient descent over an objective functional $\mathcal{F}$ greatly differ since they rely on the chosen geometry through the definitions of convexity, smoothness, or differentiability.

Wasserstein gradient descent should be thought of the analog of Riemannian gradient descent in infinite dimensions. Consequently, mirror descent yields updates on measures allowing for change of mass (see Lemma 2), while $W_2$ gradient descent preserves the mass, since the updates on measures write as pushforwards (i.e., displacement of particles supporting the measures). To summarize, the mirror descent scheme we consider is very different in nature to the gradient descent schemes based on the Wasserstein geometry (e.g. Chizat and Bach (2018); Mei et al. (2018); Rotskoff and Vanden-Eijnden (2018); Wibisono (2018); Korba et al. (2020); Salim et al. (2020); Korba et al. (2021)), due to the different geometry.

## F    Proofs

### F.1    Proof of Theorem 4

*Proof.* Since $\mathcal{F}$ is $L$-smooth relative to $\phi$ over $\mathcal{R}$ and we assumed that $(\mu_n)_{n\in\mathbb{N}} \in \mathcal{R}^{\mathbb{N}}$, we have

$$\mathcal{F}(\mu_{n+1}) \leq \mathcal{F}(\mu_n) + d^+\mathcal{F}(\mu_n)(\mu_{n+1} - \mu_n) + LD_\phi(\mu_{n+1}|\mu_n). \tag{36}$$

Applying Lemma 3 to the convex function $\mathcal{G}_n(\nu) = \frac{1}{L}d^+\mathcal{F}(\mu_n)(\nu - \mu_n)$, with $\mu = \mu_n$ and $\bar{\nu} = \mu_{n+1}$ yields

$$d^+\mathcal{F}(\mu_n)(\mu_{n+1} - \mu_n) + LD_\phi(\mu_{n+1}|\mu_n) \leq d^+\mathcal{F}(\mu_n)(\nu - \mu_n) + LD_\phi(\nu|\mu_n) - LD_\phi(\nu|\mu_{n+1}).$$

Fix $\nu \in \mathcal{R}$, then (36) becomes:

$$\mathcal{F}(\mu_{n+1}) \leq \mathcal{F}(\mu_n) + d^+\mathcal{F}(\mu_n)(\nu - \mu_n) + LD_\phi(\nu|\mu_n) - LD_\phi(\nu|\mu_{n+1}). \tag{37}$$

This shows in particular, by substituting $\nu = \mu_n$ and since $D_\phi(\nu|\mu_{n+1}) \geq 0$, that $\mathcal{F}(\mu_{n+1}) \leq \mathcal{F}(\mu_n)$, i.e. $\mathcal{F}$ is decreasing at each iteration. Since $\mathcal{F}$ is $l$-strongly convex relative to $\phi$, we also have:

$$d^+\mathcal{F}(\mu_n)(\nu - \mu_n) \leq \mathcal{F}(\nu) - \mathcal{F}(\mu_n) - lD_\phi(\nu|\mu_n) \tag{38}$$

and (37) becomes:

$$\mathcal{F}(\mu_{n+1}) \leq \mathcal{F}(\nu) + (L - l)D_\phi(\nu|\mu_n) - LD_\phi(\nu|\mu_{n+1}). \tag{39}$$

By induction, similarly to Lu et al. (2018), we sum (39) over $n$, obtaining

$$\sum_{i=1}^n \left(\frac{L}{L-l}\right)^i \mathcal{F}(\mu_i) \leq \sum_{i=1}^n \left(\frac{L}{L-l}\right)^i \mathcal{F}(\nu) + LD_\phi(\nu|\mu_0) - L\left(\frac{L}{L-l}\right)^n D_\phi(\nu|\mu_n)$$

Using the monotonicity of $(\mathcal{F}(\mu_n))_{n\geq0}$ and the positivity of $D_\phi(\nu|\mu_n)$, we have

$$\sum_{i=1}^n \left(\frac{L}{L-l}\right)^i (\mathcal{F}(\mu_n) - \mathcal{F}(\nu)) \leq LD_\phi(\nu|\mu_0) - L\left(\frac{L}{L-l}\right)^n D_\phi(\nu|\mu_n) \leq LD_\phi(\nu|\mu_0). \quad \square$$

## F.2 Proof of Proposition 5

Let $\pi, \bar{\pi} \in \mathcal{P}(\mathcal{X} \times \mathcal{Y})$, $h > 0$ and $\xi = \bar{\pi} - \pi$ hence any integral with respect to $\xi$ of constant functions is null. We have:

$$F_S(\pi + h\xi) - F_S(\pi) = \mathrm{KL}(p_\mathcal{X}(\pi + h\xi)|\bar{\mu}) - \mathrm{KL}(p_\mathcal{X}\pi|\bar{\mu})$$

$$= \int \log\left(p_\mathcal{X}\pi + hp_\mathcal{X}\xi\right) dp_\mathcal{X}(\pi + h\xi) - \int \log(\bar{\mu}) dp_\mathcal{X}(\pi + h\xi) - \int \log\left(\frac{p_\mathcal{X}\pi}{\bar{\mu}}\right) dp_\mathcal{X}\pi$$

$$= h \int \log\left(\frac{p_\mathcal{X}\pi}{\bar{\mu}}\right) dp_\mathcal{X}\xi + \underbrace{\int \log\left(1 + h\frac{p_\mathcal{X}\xi}{p_\mathcal{X}\pi}\right) dp_\mathcal{X}\pi}_{\approx h \int \frac{p_\mathcal{X}\xi}{p_\mathcal{X}\pi} dp_\mathcal{X}\pi + o(h) = 0 + o(h)} + \underbrace{h \int \log\left(1 + h\frac{p_\mathcal{X}\xi}{p_\mathcal{X}\pi}\right) dp_\mathcal{X}\xi}_{\approx h^2 \int \frac{p_\mathcal{X}\xi}{p_\mathcal{X}\pi} dp_\mathcal{X}\xi + o(h^2)}$$

$$= h \int \log\left(\frac{p_\mathcal{X}\pi}{\bar{\mu}}\right) dp_\mathcal{X}\xi + o(h).$$

Consequently,

$$\lim_{h \to 0^+} \frac{F_S(\pi + h\xi) - F_S(\pi)}{h} = \int_\mathcal{X} \log\left(\frac{p_\mathcal{X}\pi}{\bar{\mu}}\right) d\xi = \int_{\mathcal{X} \times \mathcal{Y}} \log\left(\frac{p_\mathcal{X}\pi}{\bar{\mu}}\right) d\xi. \tag{40}$$

Hence, when it exists, $\nabla_C F_S(\pi) = \ln(dp_\mathcal{X}\pi/d\bar{\mu})$. Moreover, the sets $\Pi(*, \bar{\nu})$ and $\Pi(\bar{\mu}, *)$ are $L^\infty$-weak-* closed.[6] Besides, KL has weak-* compact sublevel sets and is weak-* l.s.c. Hence Attouch et al. (2014, Theorem 3.2.2) applies, and the iterates $(\pi_n)_{n \geq 0}$ exist, as originally shown by Csiszar (1975). As $\pi_n = e^{(f+g-c)/\epsilon} \bar{\mu} \otimes \bar{\nu}$ (Nutz, 2021, Section 6) with $f \in L^\infty(\mathcal{X})$ and $g \in L^\infty(\mathcal{Y})$, we have that $x \mapsto \ln(d\mu_n(x)/d\bar{\mu}(x)) \in L^\infty(\mathcal{X}, \mathbb{R})$; indeed as $c \in L^\infty$, the first marginal $\mu_n$ of $\pi_n$ is an integral of functions bounded by strictly positive quantities.

Consider a coupling $\pi \in \mathcal{P}(\mathcal{X} \times \mathcal{Y})$ with $\pi \ll \pi_n$ and denote by $\mu$ its first marginal. We have that $F_S(\pi_n) = \int_\mathcal{X} \ln(\mu_n/\bar{\mu}) d\mu_n$ and $\langle \nabla_C F_S(\pi_n), \pi - \pi_n \rangle = \iint \ln(d\mu_n(x)/d\bar{\mu}(x))(\pi(dx, dy) - \pi_n(dx, dy))$. Simplifying and using (14) twice we obtain the identity:

$$F_S(\pi_n) + \langle \nabla_C F_S(\pi_n), \pi - \pi_n \rangle + \mathrm{KL}(\pi|\pi_n)$$

$$= \int \ln(d\mu_n/d\bar{\mu}(x))\mu_n(dx) + \iint \ln(d\mu_n/d\bar{\mu}(x))\pi(dx, dy)$$

$$\quad - \iint \ln(d\mu_n/d\bar{\mu}(x))\pi_n(dx, dy) + \mathrm{KL}(\pi|\pi_n)$$

$$= \int \ln(d\mu_n/d\bar{\mu}(x))\mu(dx) + \mathrm{KL}(p_\mathcal{X}\pi|\mu_n) + \mathrm{KL}(\pi|p_\mathcal{X}\pi \otimes \pi_n/\mu_n)$$

$$= \int \ln(d\mu/d\bar{\mu}(x))\mu(dx) + \mathrm{KL}(\pi|p_\mathcal{X}\pi \otimes \pi_n/\mu_n) = \mathrm{KL}(\pi|\bar{\mu} \otimes \pi_n/\mu_n) = \mathrm{KL}(\pi|\pi_{n+\frac{1}{2}}).$$

We conclude by taking the argmin over $\pi \in C$.

## F.3 Proof of Proposition 7

The proof of Proposition 7 essentially relies on bounding the entropic potentials by the marginals, as in Luise et al. (2019)[Theorem C.4]. For their purpose Luise et al. (2019) assume that $\mathcal{X} = \mathcal{Y}$ and that $c(x, y) = c(y, x)$. These assumptions are not needed here so we revisit their proof and show that their bound holds for general bounded costs. We define

$$D_c = \frac{1}{2} \sup[c(x, y) + c(x', y') - c(x, y') - c(x', y)],$$

where the supremum runs over $x, x' \in \mathcal{X}$ and $y, y' \in \mathcal{Y}$. When $\mu$ and $\nu$ are probability measures on $\mathcal{X}$ and $\mathcal{Y}$ respectively we define the soft-$c$ transform mappings $T_\mu \colon L^\infty(\mathcal{X}) \to L^\infty(\mathcal{Y})$ and $T_\nu \colon L^\infty(\mathcal{Y}) \to L^\infty(\mathcal{X})$ by

$$T_\mu(f)(y) = -\epsilon \ln\left(\int_\mathcal{X} e^{(f(x) - c(x,y))/\epsilon} \mu(dx)\right)$$

---

[6]Indeed, take $(\pi_n)_{n \in N} \in \Pi(*, \bar{\nu})$ converging weakly to some $\bar{\pi}$. As $\langle g, \pi_n \rangle_{\mathcal{X} \times \mathcal{Y}} = \langle g, \bar{\nu} \rangle_\mathcal{Y}$ for all $g \in L^\infty(\mathcal{Y}, \mathbb{R})$, we obtain that $\langle g, \bar{\pi} \rangle_{\mathcal{X} \times \mathcal{Y}} = \langle g, \bar{\nu} \rangle_\mathcal{Y}$ which precisely means that $p_\mathcal{Y}\bar{\pi} = \bar{\nu}$.

and
$$T_\nu(g)(x) = -\epsilon \ln \left( \int_\mathcal{Y} e^{(g(y)-c(x,y))/\epsilon} \nu(dy) \right).$$

These mappings arise naturally in the context of Sinkhorn's algorithm since if $\pi \in \Pi_c$ has marginals $(\mu, \nu)$, we can write $\pi(dx, dy) = e^{(f(x)+g(y)-c(x,y))/\epsilon}\mu(dx)\nu(dy)$ and taking marginals implies

$$g = T_\mu(f) \quad \text{and} \quad f = T_\nu(g). \tag{41}$$

Luise et al. (2019) use the *Hilbert metric* to prove their result, a classical tool to analyze matrix scaling problems (Franklin and Lorenz, 1989), which for our purpose here reduces to the following semi-norm.

**Definition 7.** When $f \in L^\infty(\mathcal{X})$ we set $\|f\|_{\text{var}} = (\sup_\mathcal{X} f) - (\inf_\mathcal{X} f)$. We similarly define $\|g\|_{\text{var}}$ for $g \in L^\infty(\mathcal{Y})$.

We are now ready to state our version of Luise et al. (2019)[Theorem C.4].

**Proposition 15.** Let $\pi, \tilde{\pi} \in \Pi_c$ with marginals $(\mu, \nu)$ and $(\tilde{\mu}, \tilde{\nu})$ respectively. Write $\pi = e^{(f+g-c)/\epsilon}\mu \otimes \nu$ and $\tilde{\pi} = e^{(\tilde{f}+\tilde{g}-c)/\epsilon}\tilde{\mu} \otimes \tilde{\nu}$. Then

$$\|f - \tilde{f}\|_{\text{var}} + \|g - \tilde{g}\|_{\text{var}} \leq 2\epsilon \, e^{3D_c/\epsilon}\big(\|\mu - \tilde{\mu}\|_{\text{TV}} + \|\nu - \tilde{\nu}\|_{\text{TV}}\big).$$

The proof of this quantitative stability estimate mainly relies on the classical result that the soft $c$-transform mappings are contractions in the Hilbert metric; this result is at the heart of the proof of the classical linear convergence rate of Sinkhorn (see Franklin and Lorenz, 1989; Chen et al., 2016, for a proof).

**Proposition 16.** $\|T_\mu(\tilde{f}) - T_\mu(f)\|_{\text{var}} \leq \lambda \|\tilde{f} - f\|_{\text{var}}$ with $\lambda = \frac{e^{D_c/\epsilon}-1}{e^{D_c/\epsilon}+1} < 1$.

We will also need the following lemma which is essentially contained in Luise et al. (2019).

**Lemma 17.** Let $f = T_\nu(g)$ for some $g \in L^\infty(\mathcal{Y})$. Then

$$\|T_{\tilde{\mu}}(f) - T_\mu(f)\|_{\text{var}} \leq 2\epsilon \, e^{2D_c/\epsilon}\|\mu - \tilde{\mu}\|_{\text{TV}}.$$

Likewise if $g = T_\mu(f)$ for some $f \in L^\infty(\mathcal{X})$,

$$\|T_{\tilde{\nu}}(g) - T_\nu(g)\|_{\text{var}} \leq 2\epsilon \, e^{2D_c/\epsilon}\|\nu - \tilde{\nu}\|_{\text{TV}}.$$

*Proof of Lemma 17.* For any $f \in L^\infty(\mathcal{X})$ we have by definition

$$T_{\tilde{\mu}}(f)(y) - T_\mu(f)(y) = \epsilon \log \left( \int_\mathcal{X} e^{(f(x)-c(x,y))/\epsilon}\mu(dx) \right) - \epsilon \log \left( \int_\mathcal{X} e^{(f(x)-c(x,y))/\epsilon}\tilde{\mu}(dx) \right).$$

To control this difference of logs, Luise et al. (2019)[Lemma C.2] use the bound $|\log(a) - \log(b)| \leq \max\{a^{-1}, b^{-1}\}|a - b|$ (for any $a, b > 0$). We have $\int_\mathcal{X} e^{(f(x)-c(x,y))/\epsilon}\mu(dx) \geq e^{\inf_x[f(x)-c(x,y)]/\epsilon}$ and the same lower bound holds for $\int_\mathcal{X} e^{(f(x)-c(x,y))/\epsilon}\tilde{\mu}(dx)$. Therefore

$$|T_{\tilde{\mu}}(f)(y) - T_\mu(f)(y)| \leq \epsilon \, e^{-\inf_x[f(x)-c(x,y)]/\epsilon} \int_\mathcal{X} e^{(f(x')-c(x',y))/\epsilon}|\mu - \tilde{\mu}|(dx')$$

$$\leq \epsilon \, e^{\sup_x[c(x,y)-f(x)]/\epsilon} e^{\sup_{x'}[f(x')-c(x',y)]/\epsilon}\|\mu - \tilde{\mu}\|_{\text{TV}}.$$

This implies when taking the supremum over $y \in \mathcal{Y}$

$$\|T_{\tilde{\mu}}(f) - T_\mu(f)\|_{\text{var}} \leq 2\|T_{\tilde{\mu}}(f) - T_\mu(f)\|_\infty \leq 2\epsilon \, e^{\sup_{x,x',y}[f(x')-f(x)+c(x,y)-c(x',y)]/\epsilon}\|\mu - \tilde{\mu}\|_{\text{TV}}.$$

This last inequality is valid for any $f \in L^\infty(\mathcal{X})$. If in addition we take $f$ to be an image $f = T_\nu(g)$, then we have the standard estimate for any given $x, x' \in \mathcal{X}$ and $y \in \mathcal{Y}$

$$-f(x) = \epsilon \ln \left( \int_\mathcal{Y} e^{(g(y')-c(x,y'))/\epsilon}\nu(dy') \right)$$

$$= \epsilon \ln \left( \int_\mathcal{Y} e^{(c(x,y)+c(x',y')-c(x',y)-c(x,y'))/\epsilon} e^{(g(y')-c(x',y'))/\epsilon}\nu(dy') \right) + c(x', y) - c(x, y)$$

$$\leq 2D_c + \epsilon \ln \left( \int_\mathcal{Y} e^{(g(y')-c(x',y'))/\epsilon}\nu(dy') \right) + c(x', y) - c(x, y)$$

$$= 2D_c - f(x') + c(x', y) - c(x, y).$$

This shows that $\sup_{x,x',y}[f(x') - f(x) + c(x,y) - c(x',y)] \le 2D_c$. As a consequence,

$$\|T_{\tilde\mu}(f) - T_\mu(f)\|_{\mathrm{var}} \le 2\epsilon\, e^{2D_c/\epsilon}\|\mu - \tilde\mu\|_{\mathrm{TV}}.$$

By symmetry the corresponding bound can be derived for quantities on $\mathcal{Y}$.  □

*Proof of Proposition 15.* Having in mind the fixed point equations (41) for $(f,g)$ and the corresponding ones for $(\tilde{f}, \tilde{g})$ we write

$$\|\tilde{f} - f\|_{\mathrm{var}} = \|T_{\tilde\nu}(\tilde{g}) - T_\nu(g)\|_{\mathrm{var}}$$
$$\le \|T_{\tilde\nu}(\tilde{g}) - T_{\tilde\nu}(g)\|_{\mathrm{var}} + \|T_{\tilde\nu}(g) - T_\nu(g)\|_{\mathrm{var}},$$

and similarly, $\|\tilde{g} - g\|_{\mathrm{var}} \le \|T_{\tilde\mu}(\tilde{f}) - T_{\tilde\mu}(f)\|_{\mathrm{var}} + \|T_{\tilde\mu}(f) - T_\mu(f)\|_{\mathrm{var}}$. By Proposition 16, $\|T_{\tilde\mu}(\tilde{f}) - T_{\tilde\mu}(f)\|_{\mathrm{var}} \le \lambda\|\tilde{f} - f\|_{\mathrm{var}}$ and $\|T_{\tilde\nu}(\tilde{g}) - T_{\tilde\nu}(g)\|_{\mathrm{var}} \le \lambda\|\tilde{g} - g\|_{\mathrm{var}}$. Combining, we obtain

$$(1 - \lambda)\big(\|\tilde{f} - f\|_{\mathrm{var}} + \|\tilde{g} - g\|_{\mathrm{var}}\big) \le \|T_{\tilde\mu}(f) - T_\mu(f)\|_{\mathrm{var}} + \|T_{\tilde\nu}(g) - T_\nu(g)\|_{\mathrm{var}}.$$

Lemma 17 takes care of the right-hand side, and this results in

$$(1 - \lambda)\big(\|\tilde{f} - f\|_{\mathrm{var}} + \|\tilde{g} - g\|_{\mathrm{var}}\big) \le 2\epsilon\, e^{2D_c/\epsilon}\big(\|\mu - \tilde\mu\|_{\mathrm{TV}} + \|\nu - \tilde\nu\|_{\mathrm{TV}}\big).$$

Finally we divide by $1 - \lambda$ and bound $(1 - \lambda)^{-1} = (e^{D_c/\epsilon} + 1)/2 \le e^{D_c/\epsilon}$.  □

*Proof of Proposition 7.* Let $\pi, \tilde\pi \in \Pi_c$ with marginals $(\mu, \bar\nu)$ and $(\tilde\mu, \bar\nu)$ respectively. Write $\pi = e^{(f+g-c)/\epsilon}\mu \otimes \bar\nu$ and $\tilde\pi = e^{(\tilde{f}+\tilde{g}-c)/\epsilon}\tilde\mu \otimes \bar\nu$. We emphasize that $\pi$ and $\tilde\pi$ have the same second marginal $\bar\nu$. Then

$$\epsilon\,\mathrm{KL}(\tilde\pi|\pi) = \epsilon\,\mathrm{KL}(\tilde\pi|\pi) + \epsilon\,\mathrm{KL}(\pi|\tilde\pi) - \epsilon\,\mathrm{KL}(\pi|\tilde\pi)$$
$$= \iint\Big(\tilde{f} - f + \tilde{g} - g + \epsilon\ln\Big(\frac{d\tilde\mu}{d\mu}\Big)\Big)d\tilde\pi + \iint\Big(f - \tilde{f} + g - \tilde{g} + \epsilon\ln\Big(\frac{d\mu}{d\tilde\mu}\Big)\Big)d\pi - \epsilon\,\mathrm{KL}(\pi|\tilde\pi)$$
$$= \iint(\tilde{f} - f + \tilde{g} - g)\,(d\tilde\pi - d\pi) + \epsilon\,\mathrm{KL}(\tilde\mu|\mu) + \epsilon\,\mathrm{KL}(\mu|\tilde\mu) - \epsilon\,\mathrm{KL}(\pi|\tilde\pi).$$

Part of the first term vanishes since $\iint(\tilde{g} - g)\,(d\tilde\pi - d\pi) = \int_{\mathcal{Y}}(\tilde{g} - g)\,(d\bar\nu - d\bar\nu) = 0$, and we can get rid of the last two terms by using the data processing inequality $\mathrm{KL}(\mu|\tilde\mu) \le \mathrm{KL}(\pi|\tilde\pi)$. Thus

$$\epsilon\,\mathrm{KL}(\tilde\pi|\pi) \le \|\tilde{f} - f\|_{\mathrm{var}}\|\tilde\mu - \mu\|_{\mathrm{TV}} + \epsilon\,\mathrm{KL}(\tilde\mu|\mu).$$

Applying Proposition 15 we obtain

$$\epsilon\,\mathrm{KL}(\tilde\pi|\pi) \le 2\epsilon e^{3D_c/\epsilon}\|\tilde\mu - \mu\|_{\mathrm{TV}}^2 + \epsilon\,\mathrm{KL}(\tilde\mu|\mu),$$

and after dividing by $\epsilon$, Pinsker's inequality yields

$$\mathrm{KL}(\tilde\pi|\pi) \le (1 + 4e^{3D_c/\epsilon})\,\mathrm{KL}(\tilde\mu|\mu).$$  □

### F.4   Proof of Proposition 9

**Proposition 18** (EM as mirror descent). Let $C = \Pi(*, \bar\nu)$. Assume that for all $\pi \in C$ there exists a $q_*(\pi) \in \mathcal{Q}$ solving (28), that, for $p_h = p_{q_*((1-h)\pi_n + h\pi)}$, $d\pi_n/dp_h$ converges pointwise to $d\pi_n/dp_{q_n}$ for $h \to 0^+$ with $|\ln(d\pi_n/dp_h)| \le \mathcal{G}_n$ for some $\mathcal{G}_n \in L^1(\pi + \pi_n)$, that $d\pi_n/dp_{q_n}(\cdot, \cdot) \in [a_n, b_n]$ for some $a_n > 0$ and $b_n > 0$, and that $\sup_{q \in \mathcal{Q}}|\ln(d\pi_n/dp_q)| < \infty$. Then the EM iterations (26)–(27) can be written as a mirror descent iteration with objective function $F_{\mathrm{EM}}$, Bregman potential $\phi_e$ and constraints $C$,

$$\pi_{n+1} = \operatorname*{argmin}_{\pi \in C}\langle\nabla_C F_{\mathrm{EM}}(\pi_n), \pi - \pi_n\rangle + \mathrm{KL}(\pi|\pi_n), \tag{42}$$

with $\nabla_C F_{\mathrm{EM}}(\pi_n) = \ln(d\pi_n/dp_{q_n}) \in L^\infty(\mathcal{X} \times \mathcal{Y})$.

**Remark 8.** Note that our assumptions on the sequence $(p_h)_{h\in[0,1]}$ in Proposition 18 are very similar to what the fundamental theorem of $\Gamma$-convergence would provide (see Dal Maso (1987), Braides (2002, Theorem 2.10)). It is indeed straightforward to prove $\Gamma$-convergence (see Braides, 2002, Theorem 2.1) of the sequence $(f_{n,\pi}(\cdot, h))_{h\in[0,1]}$ in $h = 0^+$ with $f_{n,\pi}(p, h) := \mathrm{KL}(\pi_n + h(\pi - \pi_n)|p)$, owing to the convexity and joint weak-* lower semicontinuity of KL. However, to prove the convergence of the sequence of minimizers $(p_h)_{h\in[0,1]}$, one would need the equicoercivity of $(\mathrm{KL}(\pi_h|p))_{h\in[0,1]}$ over $p \in \mathcal{P}_Q$ Braides (2002, Definition 2.9)), which heavily depends on the properties of $\mathcal{P}_Q$, e.g. considering a weak-* compact $\mathcal{P}_Q$ would entail equicoercivity.

*Proof.* We will use here the envelope theorem to differentiate $F_{\mathrm{EM}}$ and compute its first variation. We are going to apply Milgrom and Segal (2002, Theorem 3) leveraging properties of KL. Milgrom and Segal (2002, Theorem 3) is written for the set $[0, 1] \times X$, where $X$ is some set optimized over. Here $X = \mathcal{P}_Q := \{p_q \mid q \in \mathcal{Q}\}$ and the interval $[0, 1]$ will be merely the scalar of the directional derivative we consider.

Let $n \geq 0$ and $\pi \in \mathcal{P}(\mathcal{X} \times \mathcal{Y})$. For $h \in [0, 1]$, set $f_{n,\pi}(p, h) := \mathrm{KL}(\pi_n + h(\pi - \pi_n)|p)$ and $V_{n,\pi}(h) = \inf_{p \in \mathcal{P}_Q} \mathrm{KL}(\pi_n + h(\pi - \pi_n)|p)$ to match the notations of Milgrom and Segal (2002, Theorem 3). We have to show some equidifferentiability over $q \in \mathcal{P}_Q$. Notice that the following expression does not depend on $p$,

$$\frac{1}{h}\left[\iint \ln\left(\frac{\pi_n + h(\pi - \pi_n)}{p}\right) d(\pi_n + h(\pi - \pi_n)) - \iint \ln\left(\frac{\pi_n}{p}\right) d\pi_n\right] - \iint \ln\left(\frac{\pi_n}{p}\right) d(\pi - \pi_n)$$

$$= \frac{1}{h}\iint \ln\left(1 + h\frac{(\pi - \pi_n)}{\pi_n}\right) d\pi_n = \frac{1}{h}[h\iint d(\pi - \pi_n) + O(h^2)] = 0 + O(h),$$

so that we do have equidifferentiability when $h \to 0^+$. Our assumptions then allow to apply Milgrom and Segal (2002, Theorem 3). We thus obtain that

$$d^+ F_{\mathrm{EM}}(\pi_n)(\pi - \pi_n) = d^+ V_{n,\pi}(0) = \lim_{h\to 0^+} \iint \ln\left(\frac{d\pi_n}{dp_h}\right) d\pi_n.$$

Since $|\ln(d\pi_n/dp_h)| \leq \mathcal{G}_n \in L^1(\pi + \pi_n)$ and $d\pi_n/dp_h$ converges pointwise to $d\pi_n/dp_{q_n}$ for $h \to 0^+$ (recall that $q_n = q_*(\pi_n)$ by definition), we can apply the dominated convergence theorem to interchange the limit and the integral. Consequently $d^+ F_{\mathrm{EM}}(\pi_n)(\pi - \pi_n) = \iint \ln(d\pi_n/dp_{q_n}) d(\pi - \pi_n)$ proving that $\nabla F_{\mathrm{EM}}(\pi_n) = \ln(d\pi_n/dp_{q_n}) \in L^\infty$ since $\pi_n/p_{q_n}(\cdot, \cdot) \in [a_n, b_n]$ for some $a_n > 0$ and $b_n > 0$.

Then, for $\pi_n$ the EM iterate at time $n$, and for any coupling $\pi$, we have the identity:

$$F_{\mathrm{EM}}(\pi_n) + \langle \nabla_C F_{\mathrm{EM}}(\pi_n), \pi - \pi_n \rangle + \mathrm{KL}(\pi|\pi_n)$$

$$= \int \ln(d\pi_n/dp_{q_n}(x))\pi_n(dx) + \int \ln(d\pi_n/dp_{q_n}(x))(\pi - \pi_n)(dx) + \int \ln(d\pi/d\pi_n(x))\pi(dx)$$

$$= \int \ln(d\pi/dp_{q_n}(x))\pi(dx) = \mathrm{KL}(\pi|p_{q_n}).$$

Note that $q_n$ is optimal in (28), whence (29) matches (27). $\qquad\square$

### F.5 Proof of Proposition 10

*Proof.* Let $\pi, \bar{\pi} \in \mathcal{P}(\mathcal{X} \times \mathcal{Y})$, $h > 0$ and $\xi = \bar{\pi} - \pi$, so $\iint_{\mathcal{X} \times \mathcal{Y}} \xi(dx, dy) = 0$. We have:

$$F_{\text{LEM}}(\pi + h\xi) - F_{\text{LEM}}(\pi) = \text{KL}(\pi + h\xi | p_{\mathcal{X}}(\pi + h\xi) \otimes K) - \text{KL}(\pi | p_{\mathcal{X}} \pi \otimes K)$$

$$= \int \log\left(\frac{\pi + h\xi}{p_{\mathcal{X}}(\pi + h\xi) \otimes K}\right) d(\pi + h\xi) - \int \log\left(\frac{\pi}{p_{\mathcal{X}} \pi \otimes K}\right) d\pi$$

$$= \int \log(\pi + h\xi) d(\pi + h\xi) - \int \log(p_{\mathcal{X}}(\pi + h\xi) \otimes K) d(\pi + h\xi) - \int \log(\pi) d\pi + \int \log(p_{\mathcal{X}} \pi \otimes K) d\pi$$

$$= h \int \log \pi d\xi + \underbrace{\int \log\left(1 + h\frac{\xi}{\pi}\right) d\pi}_{\approx h \int \frac{\xi}{\pi} d\pi + o(h) = 0 + o(h)} + h \underbrace{\int \log\left(1 + h\frac{\xi}{\pi}\right) d\xi}_{\approx h^2 \int \frac{\xi}{\pi} d\xi + o(h^2)}$$

$$- h \int \log(p_{\mathcal{X}} \pi \otimes K) d\xi - \underbrace{\int \log\left(1 + h\frac{p_{\mathcal{X}} \xi \otimes K}{p_{\mathcal{X}} \pi \otimes K}\right) d\pi}_{\approx h \int \frac{p_{\mathcal{X}} \xi \otimes K}{p_{\mathcal{X}} \pi \otimes K} d\pi + o(h) = h \int \frac{p_{\mathcal{X}} \xi}{p_{\mathcal{X}} \pi} d\pi + o(h) = 0 + o(h)} - h \underbrace{\int \log\left(1 + h\frac{p_{\mathcal{X}} \xi \otimes K}{p_{\mathcal{X}} \pi \otimes K}\right) d\xi}_{h^2 \int \frac{p_{\mathcal{X}} \xi \otimes K}{p_{\mathcal{X}} \pi \otimes K} d\xi + o(h^2)}$$

$$= h \int \log\left(\frac{\pi}{p_{\mathcal{X}} \pi \otimes K}\right) d\xi + o(h).$$

Hence

$$\lim_{h \to 0^+} \frac{F_{\text{LEM}}(\pi + h\xi) - F_{\text{LEM}}(\pi)}{h} = \int \log\left(\frac{\pi}{p_{\mathcal{X}} \pi \otimes K}\right) d\xi.$$

To show that $\nabla F_{\text{LEM}}(\pi_n)$ belongs to $L^\infty$, we proceed by induction. Let $n \geq 0$ and assume that $T_K \mu_n \gg \bar{\nu}$ and $\mu_n = e^{f_n(x)} \bar{\mu}$ with $f_n$ bounded (which we explicitly assumed for $\mu_0$) then the multiplicative update (32) shows that $f_{n+1}$ has the same property. Furthermore (32) gives

$$\frac{\pi_{n+1}}{\mu_{n+1} \otimes K}(\cdot) = \frac{\mu_n(\cdot) \frac{K(\cdot, dy)\bar{\nu}(dy)}{\int_{\mathcal{X}} K(x, dy)\mu_n(dx)}}{\mu_n(\cdot) \otimes K(\cdot, dy) \int_{\mathcal{Y}} \frac{K(\cdot, dy')\bar{\nu}(dy')}{\int_{\mathcal{X}} K(x, dy')\mu_n(dx)}} = \frac{\bar{\nu}(dy)}{\int_{\mathcal{X}} K(x, dy)\mu_n(dx) \times \int_{\mathcal{Y}} \frac{K(\cdot, dy')\bar{\nu}(dy')}{\int_{\mathcal{X}} K(x, dy')\mu_n(dx)}}$$

Since $K(x, dy) = e^{-c(x,y)} \bar{\nu}$ with $c$ uniformly bounded, $\frac{\pi_{n+1}}{\mu_{n+1} \otimes K}(\cdot)$ is also bounded above and below by positive constants (depending on $n$). $\square$

### F.6 Proof of Proposition 11

*Proof.* By the disintegration formula (14),

$$F_{\text{LEM}}(\pi) = \text{KL}(\bar{\nu} | p_{\mathcal{Y}}(p_{\mathcal{X}} \pi \otimes K)) + \int \text{KL}(\pi/\bar{\nu} | (p_{\mathcal{X}} \pi \otimes K)/p_{\mathcal{Y}}(p_{\mathcal{X}} \pi \otimes K)) d\bar{\nu} \qquad (43)$$

Let $\pi_* = \mu_*(dx) k(x, dy) \bar{\nu}(dy)/(T_K \mu_*)(dy)$. First, for any $\pi \in \mathcal{P}(\mathcal{X} \times \mathcal{Y})$, we have $p_{\mathcal{Y}}(p_{\mathcal{X}} \pi \otimes K) = \int p_{\mathcal{X}} \pi(dx) k(x, \cdot) = T_K(p_{\mathcal{X}} \pi)$, hence by definition of $\mu^* \in \arg\min_\mu \text{KL}(\bar{\nu}|T_k \mu)$, $\pi_*$ minimizes the first term in (43). Second, this choice leads to $\pi_*/\bar{\nu} = \mu_* \otimes K/T_K \mu^*$, cancelling the nonnegative second term in (43). Hence $\pi_*$ is a minimizer of $F_{\text{LEM}}$ and $F_{\text{LEM}}(\pi_*) = \text{KL}(\bar{\nu}|T_K \mu_*)$. Moreover, $\text{KL}(\cdot|\cdot)$ is convex in both arguments, and $\pi \mapsto p_{\mathcal{X}} \pi \otimes K$ is linear. Consequently the composition $F_{\text{LEM}}$ is convex in $\pi$ and so is $F_{\text{S}}$ by the same arguments (see also Lemma 6). By (14) and linearity of the Bregman divergence, $\text{KL}(\pi|\tilde{\pi}) = D_{F_{\text{S}}}(\pi|\tilde{\pi}) + D_{F_{\text{LEM}}}(\pi|\tilde{\pi})$, hence $F_{\text{LEM}}$ is 1-relatively smooth w.r.t. $\phi_e$. Hence, Theorem 4 yields:

$$F_{\text{LEM}}(\pi_n) \leq F_{\text{LEM}}(\pi_*) + \frac{\text{KL}(\pi_*|\pi_0)}{n}.$$

Since $\pi_0 \in \Pi(*, \bar{\nu})$, $\pi_0 = \mu_0(dx) k(x, dy) \bar{\nu}(dy)/(T_K \mu_*)(dy)$,

$$\text{KL}(\pi_*|\pi_0) = \text{KL}(\mu_*|\mu_0) + \iint \ln\left(\frac{k(x, dy)\bar{\nu}(dy)/(T_K \mu_*)(dy)}{k(x, dy)\bar{\nu}(dy)/(T_K \mu_0)(dy)}\right) \pi_*(dx, dy)$$

$$= \text{KL}(\mu_*|\mu_0) + \int_{\mathcal{Y}} \ln\left(\frac{\bar{\nu}(dy)/(T_K \mu_*)(dy)}{\bar{\nu}(dy)/(T_K \mu_0)(dy)}\right) \bar{\nu}(dy) = \text{KL}(\mu_*|\mu_0) + \text{KL}(\bar{\nu}|T_K \mu_*) - \text{KL}(\bar{\nu}|T_K \mu_0).$$

Finally, we use the inequality

$$\mathrm{KL}(\bar{\nu}|T_K\mu_n) = \mathrm{KL}(\pi_n|p_{\mathcal{Y}}(p_{\mathcal{X}}\pi_n \otimes K)) \leq \mathrm{KL}(p_{\mathcal{Y}}\pi_n|p_{\mathcal{X}}\pi_n \otimes K) = F_{\mathrm{LEM}}(\pi_n).$$

$\square$