# OpenReview forum: "Mirror Descent with Relative Smoothness in Measure Spaces, with application to Sinkhorn and EM"
_NeurIPS.cc/2022/Conference — NeurIPS 2022 Accept_

### Official Review · Reviewer_Li1G · 2022-07-07

**Rating:** 6
**Confidence:** 2
**Soundness:** 4 excellent
**Presentation:** 4 excellent
**Contribution:** 2 fair

**Summary:**

When working optimizing over measures, authors analyze the infinite-dimensional mirror descent algorithm and provide O(L/t) convergence rates, when the functional is $l$ strongly convex and L-smooth, relative to another functional $\phi$, where t is the iteration count and the big O contains an initial measure of dissimilarity between the optimal measure and the initial guess. This framework is applied to optimal transport, by which using it with the KL divergence (and resulting in Sinkhorn's algorithm) and convergence results are recovered. Expectation maximization is also shown to be equivalent to the mirror descent in their framework and it is shown that one particular case is convex and thus their framework applies and guarantees sublinear convergence. Part of the contribution lies in technical generalizations of the operations and analysis to the infinite-dimensional case via the use of directional derivatives as opposed to assuming a setting in which derivatives of some kind (Gateaux, Frechet...) exist.

**Questions:**

I would have liked to see more discussion on the motivation of the problem, in comparison to what has been studied. The results applying to Sinkhorn for optimal transport were already known, am I right? It is nice that the framework allows to recover this, but I want to understand whether there was some generalization made with respect to assumptions or any other thing in this problem.

Also, was there any previous analysis for Richardson-Lucy deconvolution yielding convergence rates?

Regarding the motivating problem mentioned in the introduction, in which the functional is identified to an MMD, and regarding the implications of the results of this work to this setting, the relative smoothness 4 c_k in line 587 seems to be a very large quantity. Can you quantify / are there quantifications of this value for the application mentioned regading the infinite-width one hidden layer neural network?



**Limitations:**

yes

**Strengths And Weaknesses:**

I am not an expert in the topic of measure spaces and infinite dimensional analysis of mirror descent algorithms or other algorithms. For this reason, my evaluation of this work is somewhat limited.

Strengths:

+ Good discussion of related work, when justifying where some ideas come from in the analysis or when some theorems' proofs are inspired by others. Still, I would have liked to see a couple of comments regarding related work, see the questions section.

+ The convergence rates for Richardson-Lucy deconvolution.

+ General framework of mirror descent with its convergence rates (Theorem 3)

Weaknesses:

+ See the questions section.


Minor:

L119 As a direct consequence of Lemma 11 in *the* Appendix
L184 "which proof" -> "whose proof"

---

> ### Author Response · Authors · 2022-08-02
> **Detailed answer to Reviewer Li1G**
>
> We thank the reviewer his positive comments and interest.
>
> Question 1.: As written in the end of Section 4.1, "The linear convergence of Sinkhorn for bounded costs c has been known since at least Franklin and Lorenz (1989) and has then been derived also in the non-discrete case and in multimarginal settings (see Carlier, 2022, and references therein). Léger (2020) first obtained sublinear rates for
> unbounded costs leveraging relative smoothness, using (12) formally and through dual iterations
> on the potentials. Here we derive the same rate rigorously with a more direct proof using primal
> iterations, and complete the picture by recovering linear rates of convergence.'' In all of Section 4.1, we assume that the cost is bounded (in $L^{\infty}$) to manipulate well-defined quantities (first variations in $L^\infty$). Using the primal/measure rather than a dual/potential viewpoint makes for an easier proof of the convergence of Sinkhorn. You may also be interested by our answer to Reviewer rqqG on this topic.
>
> Question 2. To the best of our knowledge, the rates we obtain for Lucy-Richardson are novel.
>
>
> Question 3. "About the quantity $c_k=\sup_{x}k(x,x)$ and one hidden layer neural networks''. Thank your for the interesting question. Consider a regression task where the labelled data $(z,y)$ is distributed according to $P$ some data distribution. For any input $z$, the output of a single hidden layer neural network parametrized by $w$ writes as:
> \begin{equation*}
>     f_w(z)=\frac{1}{N}\sum_{j=1}^N a_j \sigma(\langle b_j, z\rangle) = \frac{1}{N}\sum_{j=1}^N \phi(z,w_j) \to \int \phi(z,w)d\mu(w) \text{ as }N \to \infty,
> \end{equation*}
> where $a_j$ and $b_j$ denote output and input weights of neuron $j=1,\dots,N$ respectively and $w_j=(a_j,b_j)$, see the references cited l24-27. In the infinite-width setting, the limiting risk is the MSE written for any $\mu$ as $\E_{(z,y)\sim P}[\|y - \int \phi(z,w)d\mu(w)\|^2]$.   When the model is well-posed, i.e. there exists a distribution $\mu^*$ over weights such that $\mathbb{E}[Y|Z=z]=\int\phi(z,w)d\mu^*(w)$, then  the limiting risk writes as an MMD with $k(w,w')= \mathbb{E}_{z\sim P}[\phi(z,w)^T \phi(z,w)]$ (see [Arbel et al 2020], Prop 20 in Appendix F). Hence, the bound on  $c_k$ depends on the choice of the activation function $\sigma$ and the output weights. If $\sigma$ is bounded (e.g. $\sigma$ is the sigmoid) then the bound on $c_k$ is the bound on the output weights. If $\sigma$ is relU, then the bound depends on the bound on input/output weights as well as on the data distribution $P$.
>
> Arbel et al. (2019). Maximum mean discrepancy gradient flow (arXiv:1906.04370)

---

### Official Review · Reviewer_SWHV · 2022-07-07

**Rating:** 6
**Confidence:** 4
**Soundness:** 3 good
**Presentation:** 3 good
**Contribution:** 3 good

**Summary:**

This is a mathematical paper on calculus of variation. The main contribution is an extension of the paper by Lu et al where in finite dimensional setting authors introduced notion of relative smoothness and relative convexity using notion of Bergman divergence and proposed corresponding Mirror Descent algorithm for which linear convergence holds. The extension to infinite dimensional space of measures is not trivial and there are a few different notions of differentiability that one may consider. Authors work with generic  Gâteaux and Fréchet derivatives which are then used to define Bergman divergence.  As an application of the theory they recover linear convergence rate for Sinkhorn algorithm in the case when the cost function c is bounded and a special setting for abstract EM algorithm.

**Questions:**

- In Lu et al. (Which I was not aware until now) Authors did a descent job presenting in which situations one cannot except to prove convergence of classical mirror descent but the modification using notions of relative smoothness and convexity fixed the problem.  In the current submissions a thorough discussion is missing.
 - One popular example of optimising functions of a measure is the story of one-hidden layer neural network in mean filed regime (works of Montanari et al. Chizat and Bach, Hu et al.). Most of the works use W_2 gradient flows perspective. It would be interesting to shed some light how ideas developed by authors could help with establishing exponential convergence in that setting.

**Ethics Review Area:**

["I don’t know"]

**Limitations:**

The work is purely theoretical and hence will not have negative social impact.

**Strengths And Weaknesses:**

Strengths:
- The paper is carefully written with the theoretical parts well flesh out.
- Adaptation of finite dimensional optimisation techniques to infinite spaces of measures allow for elegant analysis of many of the machine learning algorithms and so is of value.
-  Connection between Sinkhorn algorithm and Mirror descent while not entirely new is interesting

Weaknesses:
- The paper is purely mathetmical and I’m not convinced that NeurIPS is the best venue for this type of work. I believe mathematical journal on (variational) analysis would be more appropriate.
-  There is a large body work on differential and sub differential calculus on the spaces of measures (e.g book by Ambrosio, Luigi and Gigli, Nicola and Savaré, Giuseppe) and differentiation along the gradient flow as first proposed by Otto and Otto In Villani in their seminar paper about HWI inequality e.g W_2 or/and Fisher-Rao gradient flows . I would appreciate more through discussion  and comparisons to these works. Though I suspect that the main results would remain true for these other notions (modulo technical assumptions)

---

> ### Author Response · Authors · 2022-08-02
> **Detailed answer to Reviewer SWHV**
>
> We acknowledge that the paper is mathematical, however mirror descent with relative smoothness/convexity is an important theme in optimization and machine learning, and the Sinkhorn and EM algorithm are of great interest to the ML community. We thought it was more appropriate to confront our results to the ML community.
>
>
> Concerning the link between our framework and the one given by the Wasserstein-2 ($W_2$) geometry, developped by Ambrosio, Gigli, Savaré and Otto, Villani, we are reasonably familiar with these references as well. Given an optimisation problem over the set of probability distributions over $\R^d$, one can consider at least two frameworks. The one adopted in our paper casts the space of probability distributions as a subset of a normed space measures such as $L^2$ or Radon. The shortest distance paths between measures are given by their square-norm distance. One can consider the duality of measures with continuous functions and the mirror descent algorithm, as we do in this work. In contrast, the second framework given by $W_2$ geometry,  restricts the search space to probability distributions with bounded second moments. Equipped with the $W_2$ distance, this space is a metric space equipped with a rich Riemannian structure (often referred to as "Otto calculus") where the shortest distance paths are given by the $W_2$ distance and associated geodesics. One can leverage the Riemannian structure to discretize ($W_2$) gradient flows and consider algorithms such as ($W_2$) gradient descent.
> While both frameworks yield optimisation algorithms on measure spaces, the geometries and algorithms are quite different. The notion of convexity differs (along $L^2$ versus $W_2$ geodesics), as well as gradients (first variation vs gradient of first variation) and consequently many definitions. In addition, mirror descent yields multiplicative updates on measures allowing for change of mass, while gradient descent corresponds to displacement of (fixed mass) particles supporting the measures. A third option, Fisher-Rao gradient flows, is closer in spirit to what we study here, as the space of measures is equipped with the Hellinger distance, whose geometry is similar to the one we consider and allows for local change of mass and discontinuous displacement of particles.
>
>
> Question 1: relative smoothness is particularly well illustrated by the case where the objective function is a Kullback-Leibler divergence $\KL(\mu|\pi)$ (as in our examples Section 4). The latter is always relatively smooth to itself (hence to the KL Bregman divergence), a fact that we exploit extensively in Section 4. However the KL is typically not a smooth objective in the classical sense, i.e. with "Lipschitz gradients", as defined in Lemma 3.1 of [Chizat 2021]. Indeed the "gradient of the KL" $\mu\mapsto\log(\mu|\pi)(.)$ typically does not belong to $L^{\infty}$, in contrast to the functional $\mu\mapsto \int \Phi d\mu$ as in [Chizat 2021] for given $\Phi\in L^\infty$. A more direct argument is that traditional smoothness cannot hold because $\KL$ diverges for Dirac masses, thus does not have subquadratic growth with respect to any norm on measures. This makes $\KL$ an unsuitable objective for traditional analysis despite being ubiquitous, and indeed "relative smoothness indeed fixes the problem''.
>
> Chizat (2021) Convergence rates of gradient methods for convex optimization in the space of measures. (arXiv:2105.08368)
>
> Question 2. Thank you for the interesting question about the relation of our work to the line of work you mention, which we also cite l24-27. The latter studies (stochastic) gradient descent as a time-discretization of some continuous dynamics corresponding to the $W_2$ gradient flow (WGF) of some objective functional (a limit risk) minimized by infinite-width neural networks. However, the mirror descent scheme we consider is very different in nature to the gradient descent scheme considered in these works, due to the different geometries at stake described earlier in this answer. Another way to see it is that (Wasserstein) gradient descent can be written as proximal iterations, similarly to mirror descent (see Eq 7 in our Section 3), but with the difference that the Wasserstein distance is not a Bregman divergence. Hence, the conditions needed for convergence greatly differ in these two settings for the mirror descent versus gradient descent schemes. While we can obtain global convergence of mirror descent thanks to the convexity and smoothness of the risk in our geometry (the limit risk takes the form of an MMD, which is relatively smooth with respect to the KL, see our Prop 13); the same risk is typically not convex with respect to the Wasserstein geometry (see Prop 5 in [Arbel et al 2020] for instance), whence exponential convergence cannot be shown through this argument.
>
> Arbel et al. (2019). Maximum mean discrepancy gradient flow (arXiv:1906.04370)

---

### Official Review · Reviewer_rqqG · 2022-07-11

**Rating:** 9
**Confidence:** 4
**Soundness:** 4 excellent
**Presentation:** 4 excellent
**Contribution:** 4 excellent

**Summary:**

The paper studies mirror descent on measure spaces under relative smoothness and relative convexity assumptions. This setting is a significant generalisation of the classical $\mathbb{R}^d$ setting and basic definitions such as the definition of Bregman divergence needs to be reworked (using directional derivatives). The main results are similar convergence results as in the classical setting (such as Lu et al. 2018). As application, explicit convergence rates are obtained for the Sinkhorn and the latent EM algorithms.

**Questions:**

In Proposition 6, you define the quantity $D_c$ and explain how is it related to the relative strong convexity property of F_S.
In practice, this is only finite if the cost function c is bounded.
In Proposition 7 equation (22), it is not entirely clear whether the second bound also holds for unbounded cost functions (i.e. this was not an assumption), would you be able to state this explicitly?

It is not clear whether the sub-linear bound of O(1/n) dependence is the best we can hope for unbounded cost functions. Could the authors include an example or a reference to illustrate what happens in this case?

**Limitations:**

The authors have adequality addressed the limitations of their work.

**Strengths And Weaknesses:**

The paper introduces a beautiful new theory of relative smoothness and convexity of functionals acting on measure spaces. The study of mirror descent on measure spaces has been gaining interest in recent years, and this paper offers a significant contribution to the understanding of this method. It is quite remarkable that two widely used algorithm, the Sinkhorn and latent EM algorithm are actually special cases of the mirror descent algorithm, showing the generality of the results.

Extending the theory of relative smoothness and convexity to the measure space setting was not at all straightforward, and required a significant amount of work, including the change of some basic definitions.

Since both the theory and the applications of the paper are very convincing, we do not feel that there are any notable weaknesses.

---

> ### Author Response · Authors · 2022-08-02
> **Detailed answer to Reviewer rqqG**
>
> We thank the reviewer for his very positive and encouraging comments, as well as for acknowledging our work on the non-trivial extension of the notions of relative smoothness  and convexity to measure spaces.
>
> "In Proposition 6, you define the quantity $D_c$ and explain how is it related to the relative strong convexity property of $F_S$. In practice, this is only finite if the cost function c is bounded. In Proposition 7 equation (22), it is not entirely clear whether the second bound also holds for unbounded cost functions (i.e. this was not an assumption), would you be able to state this explicitly?"  In all of Section 4.1 we assume that the cost is bounded (in $L^{\infty}$). This indeed guarantees that $D_c$ in finite in Proposition 6. Then Prop. 6 allows us to obtain the first inequality in bound (22) in Prop 7. The other inequality of (22),
> \[\KL(\mu_n|\mu_*) \le \frac{\KL(\pi_*|\pi_0)}{ n}\]
> is inherited from relative smoothness (see Lemma 5) which comes from direct computations. We do know from [Leger 2020, Nutz 2021] that this bound holds for unbounded costs. However in the context of our general mirror descent framework and in order to manipulate finite quantities in the computations we needed to restrict ourselves to bounded costs in our article.
>
> "It is not clear whether the sub-linear bound of $O(1/n)$ dependence is the best we can hope for unbounded cost functions. Could the authors include an example or a reference to illustrate what happens in this case?"
> Let us first say that the tightness of this bound is an open question in the OT community. We will now write what is known up to our knowledge. First, $(KL(\mu_n|\mu_*))n$ is a decreasing summable nonnegative sequence and as such satisfies $\KL(\mu_n|\mu_*) = o(1/n)$ (see for instance Lemma 6.11 from [Nutz 2021]), so we have a $o(1/n)$ instead of a $O(1/n)$. But the $O(1/n)$ has here an explicit constant so it is often preferred in practice. Second, we know of simple examples with infinite costs that can be computed explicitly and that are $O(1/n^2)$. Here is one of them: take $\X=\Y=\{0,1\}$ a set with two elements and $c$ given by $c(0,0)=c(0,1)=c(1,1)=0$ and $c(1,0)=\infty$. Take $\mu=\nu=(\frac12,\frac12)$ the  uniform measure. Then there exists a solution $\pi_*$ and Sinkhorn produces iterates satisfying $\KL(\mu_n|\mu_*)\sim \frac{1}{n^2}$. It is an open question whether $O(1/n^2)$ always holds for unbounded or infinite costs.

---

> > ### Comment · Reviewer_rqqG · 2022-08-07
> > **Acknowledgement of rebuttal answers**
> >
> > We thank the authors for answering our questions.  It would be nice if the authors could include this discussion about the rates of convergence in the paper or in the supplementary material. We keep our rating unchanged.

---

### Official Review · Reviewer_w1zg · 2022-07-15

**Rating:** 6
**Confidence:** 3
**Soundness:** 3 good
**Presentation:** 3 good
**Contribution:** 2 fair

**Summary:**

The submission presents an infinite dimensional extension of relative measures of smoothness and strong-convexity to handle non-Euclidean notions of regularity. The theoretical guarantees cover the finite dimensional results recently obtained for Sinkhorn's algorithm in optimal transport and the Expectation Maximization algorithm for probabilistic models with latent variables.


**Questions:**

(Minor points)

> 36: When using the KL divergence as Bregman divergence, mirror descent yields multiplicative updates, such as Sinkhorn’s algorithm

This statement might benefit from a qualification to the KL divergence "on discrete distributions" or measures. This is to avoid the confusion that "mirror descent + KL divergence = MWU", as is can also model divergences between parametrized measures (as the KL divergences between Gaussians with fixed identity covariance leads to Euclidean gradient descent on the parameters).


**Limitations:**

No concerns

**Strengths And Weaknesses:**

The submission is well presented and well contextualized within the line of work on relative smoothness and its application to probabilistic models. My only major concern is the approachability of the submission to an audience familiar with the optimization literature, including the recent developments in relative smoothness and its application to probabilistic models, but unfamiliar with measure spaces. An exhaustive introduction to the mathematical background is of course out of scope, and I appreciated the references the submission already provides. But the motivation and significance of the presented work could be made clearer by a concrete example, not covered by prior work, where the presented framework applies. The links with Optimal Transport and EM are beneficial, but abstract. An instantiation of an infinite dimensional transport problem or a probabilistic model with infinite dimensional latent variables, to serve as equivalents of the toy problems of transporting an histogram or a mixture model of 2 Gaussians.

---

> ### Author Response · Authors · 2022-08-02
> **Detailed answer to Reviewer w1zg**
>
> "the motivation and significance of the presented work could be made clearer by a concrete example'':  Let us recall that in the paper we develop in a generic way the theory for Sinkorn and Latent EM. As described in Section 4.2, the general goal of EM is to fit, through the objective function $F_{\text{EM}}$, a parametric distribution to some observed data $Y$ (e.g. a mixture of Gaussians approximating the data), where one needs to estimate both the latent variable distribution on $X$  (e.g. weights of each Gaussian) and parameters of conditionals $P(Y|X=x)$ (e.g. means and covariances of each Gaussian). Latent EM (and its associated objective $F_{\text{LEM}}$) focuses on learning the mixture weights, since it consists in optimizing over the nonparametric latent distribution, which can be continuous or discrete. A more concrete or familiar example in machine learning  is the following: taking a discrete latent distribution $\mu$ supported on $\{1,\dots, N\}$, the goal is to learn the weights of $N$ Gaussians fitting the data distribution $\bar \nu$.
>
> Taking the limit of $N\rightarrow \infty$, we obtain general deblurring problems where the goal is to deblur a signal $Y$ given a filter $K$ (which cause a blur) and one aims at recovering the latent distribution of states $X$.
> This is further emphasized by the fact that "Latent EM'' iterations correspond to those of Lucy-Richardson algorithm, a commonly used denoising tool. Replace $\R^d$ with any Hilbert space, for instance of signals or trajectories, and you obtain a problem of finding the best distribution of input signals matching a distribution of output observations for known filter $K$.
> Such problems and schemes have already attracted a lot of interest in the statistics and signal literature previously, but the analysis of these schemes through relative smoothness and the results we obtain are novel to the best of our knowledge.
>
>
> "This statement might benefit from a qualification to the KL divergence "on discrete distributions" or measures.'': Indeed, the multiplicative updates only occur in measure space of the parameters, not in the parameters themselves (as shown with the EM algorithm). We agree with your remark - we will make this point clearer: "mirror descent yields multiplicative updates in the space of measures''

---

### Author Response · Authors · 2022-08-02
**General answer**

We thank the reviewers for their interest, positive comments and their relevant suggestions. We replied to each reviewer in a dedicated answer and will incorporate these points into the text. We hope our clarifications answer their questions and will improve their confidence and ratings concerning the paper.

---

### Meta-Review · Area_Chair_yGT2 · 2022-08-22

**Recommendation:** Accept
**Confidence:** Certain

**Metareview:**

All reviewers recommend the paper. The authors should think about ways to make the paper more accessible to a machine learning audience, but I recommend accepting. When preparing the camera-ready version, please take into account the reviewers comments and please also specifically address these two points raised in the discussion:

"I'm of the opinion that authors should try put more effort in making current submission more accessible to general audience helping the reader to understand why certain notions of differentiability have been chosen over others etc."

"Providing a concrete example where relative smoothness fails but the proposed approach applies would increase the potential audience among non-experts."

**Award:**

No

---

### Decision · Program_Chairs · 2022-09-14

Accept